# Enhancing Zero-Shot Black-Box Optimization via Efficient Population Modeling, Interaction, and Stable Gradient Approximation

**Muqi Han**
Guangzhou Institute of Technology
Xidian University
Key Laboratory of Data and Intelligent System Security Ministry of Education
`mqhan@stu.xidian.edu.cn`

**Xiaobin Li**
School of Artificial Intelligence
Xidian Univeristy
`22171214784@stu.xidian.edu.cn`

**Kai Wu**[*]
School of Artificial Intelligence
Xidian Univeristy
`kwu@xidian.edu.cn`

**Xiaoyu Zhang**
School of Cyber Engineering
Xidian University
`xiaoyuzhang@xidian.edu.cn`

**Handing Wang**
School of Artificial Intelligence
Xidian Univeristy
`hdwang@xidian.edu.cn`

## Abstract

Zero-shot optimization aims to achieve both generalization and performance gains on solving previously unseen black-box optimization problems over SOTA methods without task-specific tuning. Pre-trained optimization models (POMs) address this challenge by learning a general mapping from task features to optimization strategies, enabling direct deployment on new tasks.

In this paper, we identify three essential components that determine the effectiveness of POMs: (1) task feature modeling, which captures structural properties of optimization problems; (2) optimization strategy representation, which defines how new candidate solutions are generated; and (3) the feature-to-strategy mapping mechanism learned during pre-training. However, existing POMs often suffer from weak feature representations, rigid strategy modeling, and unstable training.

To address these limitations, we propose EPOM, an enhanced framework for pre-trained optimization. EPOM enriches task representations using a cross-attention-based tokenizer, improves strategy diversity through deformable attention, and stabilizes training by replacing non-differentiable operations with a differentiable crossover mechanism. Together, these enhancements yield better generalization, faster convergence, and more reliable performance in zero-shot black-box optimization.

## 1 Introduction

Black-box optimization (BBO) problems are ubiquitous in machine learning and engineering. The characteristic of BBO is that the algorithm can evaluate $f(\mathbf{x})$ for any solution $\mathbf{x}$; however, access to

---

[*]Corresponding author

39th Conference on Neural Information Processing Systems (NeurIPS 2025).

additional information about $f$, such as Hessian matrices and gradients, is unavailable. BBO problems include hyperparameter optimization (HPO) [1], neuroevolution [2, 3, 4], neural architecture search (NAS) [5], and algorithm selection [6], etc.

Traditionally, solving BBO tasks requires expert-designed or heavily tuned optimization algorithms for each new problem, making the process inefficient and non-scalable. Thus, the paradigm of solving new optimization problems without any task-specific tuning is extremely valuable.

Meta-learning, or learning to learn [7], particularly in the context of meta-black-box optimization [8, 9] or learning to optimize [10], covers a wide range of scenarios. It improves the performance of the optimizer on the target task by pre-training it on similar tasks. However, they typically exhibit poor generalization to new and unseen tasks. Although some methods such as LES [11] and LGA [12] can generalize to novel settings, their performance still falls significantly short compared to state-of-the-art optimizers specifically designed for these settings.

To better highlight our contribution, we introduce the concept of zero-shot optimization, which aims to simultaneously achieve strong generalization and high performance on new optimization problems without requiring task-specific training. A particularly promising direction in this area is the development of pre-trained optimization models (POMs) [13]. These models are designed to learn general-purpose optimization behaviors in a wide variety of training tasks and effectively transfer that knowledge to previously unseen problems.

We view a POM as a function that maps task-specific features to optimization strategies. This process involves three key components: (1) **Task feature modeling**, which captures characteristics of the problem (e.g. normalized fitness values and centralized rankings); (2) **Optimization strategy representation**, which determines how candidate solutions are generated and refined; (3) **Mapping mechanism**, which learns the transformation from features to strategies during pre-training.

Although recent POMs have made progress, they remain limited in three areas:

- The expressiveness of task features is insufficient to generalize across diverse problem structures;
- The strategy generation process lacks flexibility and diversity;
- The training process suffers from gradient instability due to non-differentiable operations.

To overcome these limitations, we propose **EPOM (Enhanced Pretrained Optimization Model)**, which systematically enhances all three components:

- **Feature modeling** is improved by a cross-attention-based tokenizer that captures decision-variable-level information and encodes it into fixed-length task representations;
- **Strategy representation** is enhanced through deformable attention, enabling dynamic and diversity-aware interactions among population;
- **Training stability** is ensured by replacing sampling-based crossover operations with a differentiable, weighted-sum mechanism, leading to smoother gradient flow.

By jointly improving feature modeling, strategy generation, and training robustness, EPOM achieves superior generalization and performance in zero-shot black-box optimization. This work contributes to the growing body of meta-learning literature on learning-to-optimize methods, and offers a scalable pathway toward universal optimization agents.

## 2   Related Work

**Manually Designed Population-Based BBO Algorithms.** Traditional population-based black-box optimization (BBO) algorithms, such as genetic algorithms [14], evolution strategies [15], particle swarm optimization [16], and differential evolution [17], have been widely used to tackle optimization challenges. State-of-the-art methods like CMA-ES [18] and L-SHADE [19] rely heavily on expert knowledge and trial-and-error development, resulting in limited flexibility, high development costs, and suboptimal generalization [20].

**Meta-Learned Population-Based BBO Algorithms.** Meta-learned BBO algorithms improve generalizability through meta-learning, can be categorized by their training strategies. One category

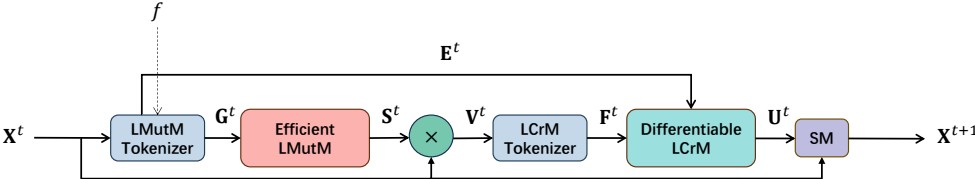

Figure 1: The overall framework of EPOM. The LMutM Tokenizer and LCrM Tokenizer are designed to generate input tokens for the Efficient LMutM and Differentiable LCrM, which are responsible for obtaining the mutant population $\mathbf{V}^t$ and the candidate population $\mathbf{U}^t$, respectively. Finally, the SM produces the offspring population based on $\mathbf{X}^t$ and $\mathbf{U}^t$.

employs bi-level optimization [21], as seen in [22], which adapts optimizers to specific task classes. LES [11] and LGA [12] use neuroevolution with self-attention mechanisms, but their reliance on external search algorithms such as OPENAI-ES [23] leads to slow training and limited scalability. Meta-MOGA [24] addresses multi-objective BBO problems by parameterizing mutation and crossover operators with multi-head self-attention and the selection operator with an MLP, whose parameters are optimized via an evolutionary algorithm. Another adopts reinforcement learning (RL) [25], for example, Shala et al. [26] use RL to meta-learn policies for adjusting CMA-ES parameters, however, suffers from training instability. Moreover, Q-Mamba [4] offline learns a meta-level policy for dynamic algorithm configuration over an optimization task distribution, while ConfigX [27] meta-learns a universal configuration policy over a modular evolutionary algorithm space. Alternatively, EvoTF [28] incorporates algorithm distillation, but requires large-scale datasets and struggles to surpass the performance of its source algorithms. Besides these, B2Opt [29] and POM [13] leverage self-supervised learning, however, B2Opt is restricted by iteration limits, and POM faces challenges in scaling with model size.

**LLMs for Optimization.** In the field of optimization, a series of optimization methods based on Large Language Models (LLMs) have emerged. They are widely applied in areas such as NP-hard problems [30, 31], algorithm evolution [32, 33, 34, 35, 36], reward design [37], and Neural Architecture Search (NAS) [38, 39]. LLMs play a crucial role in optimization, especially in sampling new solutions. However, as stated in [40], their optimization strategies rely on externally-introduced natural selection mechanisms, limiting their effectiveness in numerical optimization scenarios. For example, LLaMoCo [41] and EoH [42] use LLMs to generate code for optimization problems. The former depends on well-designed instructions and prompts, while the latter faces high evaluation costs, restricting practical applications. Moreover, TNPs [43], ExPT [44], and LICO [45] use transformer structures to address the BBO problem. Nevertheless, TNPs relies on the context information of the target problem, and neither ExPT nor LICO can be directly applied to tasks with different dimensions from the training ones. In summary, although LLM-based optimization methods have made progress, compared with pre-trained BBO methods, they have insufficient cross-task solution-generation capabilities and poor generalizability, which remain key issues to be overcome in the optimization field.

## 3 EPOM

A black-box optimization problem can be transformed as a minimization problem, and constraints may exist for corresponding solutions: $\min_{\mathbf{x}} f(\mathbf{x}), s.t. \ x_i \in [l_i, u_i]$, where $\mathbf{x} = (x_1, x_2, \cdots, x_d)$ represents the solution of optimization problem $f$, the lower and upper bounds $\mathbf{l} = (l_1, l_2, \cdots, l_d)$ and $\mathbf{u} = (u_1, u_2, \cdots, u_d)$, and $d$ is the dimension of $\mathbf{x}$. A population consists of $n$ individuals, denoted as $\mathbf{X} = [\mathbf{x}_1, \mathbf{x}_2, \cdots, \mathbf{x}_n]^{\mathrm{T}}$.

### 3.1 Review of POM

#### 3.1.1 Main Parts

The POM primarily consists of three modules: Learned Mutation Module (**LMutM**), Learned Crossover Module (**LCrM**) [2], and Selection Module (**SM**).

**1) LMutM.** LMutM generates the mutant population $\mathbf{V}^t$ through the multi-head self-attention mechanism [46]:

$$\mathbf{S}^t \leftarrow \text{LMutM}(\mathbf{H}^t), \quad \mathbf{V}^t = \mathbf{S}^t \times \mathbf{X}^t, \tag{1}$$

where $\mathbf{V}^t, \mathbf{X}^t \in \mathbb{R}^{n \times d}$, $\mathbf{H}^t \in \mathbb{R}^{n \times 2}$ and $\mathbf{S}^t \in \mathbb{R}^{n \times n}$. Here, n denotes the population size and d denotes the problem dimension. $\mathbf{X}^t$ represents the parent population, $\mathbf{H}^t = [\mathbf{h}_1^t, \mathbf{h}_2^t, \ldots, \mathbf{h}_n^t]$ is obtained by tokenizing $\mathbf{X}^t$ and serves as the input to LMutM, where each token $\mathbf{h}_i^t$ encodes the mutation features of the corresponding individual $\mathbf{x}_i^t$, including: 1) $\hat{f}_i^t$: the normalized fitness $f(\mathbf{x}_i^t)$ of $\mathbf{x}_i^t$; 2) $\hat{r}_i^t$: the centralized ranking of $\mathbf{x}_i^t$. $\mathbf{S}^t$ is the weight matrix computed by the attention mechanism.

**2) LCrM.** LCrM generates the crossover rate $\mathbf{cr}^t = [cr_1^t, cr_2^t, \cdots, cr_n^t]$ ( $\mathbf{cr}^t \in \mathbb{R}^{1 \times n}$, $cr_i^t$ is the crossover rate of the $i$-th individual) by an FFN [46] and performs the crossover operation to obtain the candidate population $\mathbf{U}^t \in \mathbb{R}^{n \times d}$:

$$\mathbf{cr}^t \leftarrow \text{LcrM}(\mathbf{Z}^t), \ \mathbf{U}^t = gumbel\text{-}softmax([\mathbf{X}^t, \mathbf{V}^t], \mathbf{cr}^t) \tag{2}$$

Where $\mathbf{Z}^t = [\mathbf{z}_1^t, \mathbf{z}_2^t, \cdots, \mathbf{z}_n^t] \in \mathbb{R}^{n \times 3}$ is the crossover feature matrix utilized by LCrM. Each token $\mathbf{z}_i^t$ encodes the crossover features of the $i$-th candidate individual $\mathbf{u}_i^t$, including: 1) $\hat{f}_i^t$: the normalized fitness $f(\mathbf{x}_i^t)$ of $\mathbf{x}_i^t$; 2) $\hat{r}_i^t$: the centralized ranking of $\mathbf{x}_i^t$; 3) $sim_i^t$: the cosine similarity between $x_i^t$ and $v_i^t$. The $gumbel\text{-}softmax$ trick [47], which independently operates across dimensions by taking $[x_{i,j}^t, v_{i,j}^t]$ ($j = 1, 2, \ldots, d$) as two categories and $[cr_i^t, 1 - cr_i^t]$ as their respective probabilities, provides a differentiable approximation of the crossover operation.

**3) SM.** Finally, SM [48], a *1-to-1* selection strategy is executed between $\mathbf{U}^t$ and $\mathbf{X}^t$ to produce the next-generation population $\mathbf{X}^{t+1}$, where $\mathbf{X}^{t+1} = \text{SM}(\mathbf{X}^t, \mathbf{U}^t)$.

#### 3.1.2 Issues

**Issues with LMutM**. 1) Insufficient and non-diverse strategies: The mutation strategies are characterized by the weight matrix $\mathbf{S}^t$ derived from the self-attention mechanism, where each individual interacts with the entire population. However, such global interaction can hinder the efficiency and diversity of the mutation process. 2) Training instability: The mask operation randomly masks parts of $\mathbf{S}^t$, which may obscure significant signals, thereby slowing down convergence and causing training instability. 3) Insufficient input features: The input $\mathbf{H}^t$ of LMutM only includes fitness and ranking information, overlooking the distribution characteristics of the population. This limits the POM's convergence and generalization abilities.

**Issues with LCrM**. LCrM uses the gumbel softmax trick to address the non-differentiability of crossover operation. However, the imprecise gradient approximation can lead to instability during model training.

### 3.2 Proposed EPOM

Our goal is to improve the strategic efficiency and trainability of POM. We have redesigned the LMutM and LCrM of POM and proposed the EPOM, as shown in Fig. 1.

#### 3.2.1 Overall Architecture

First, the LMutM Tokenizer encodes $\mathbf{X}^t$ into fixed-dimensional tokens $\mathbf{E}^t$ using a cross-attention mechanism [46], similar in spirit to prior cross-attention-based tokenization methods [49]. These tokens are then concatenated with the tokens $\mathbf{H}^t$ from the original LMutM in POM to form $\mathbf{G}^t$.

---

[2]To avoid potential confusion between the abbreviations of *Learned Mutation Module* and *Learned Crossover Module* (formerly LMM and LCM) with *Large Multimodal Models* and *Large Context Models*, we have revised their abbreviations to *LMutM* and *LCrM*, respectively.

Subsequently, the Efficient LMutM takes $\mathbf{G}^t$ as input and generates the mutant population $\mathbf{V}^t$ through two self-attention mechanisms. The first self-attention mechanism identifies, for each parent individual, the subset of population members it should attend to. The second mechanism, a deformable self-attention module [50], computes the attention weight vector $\mathbf{s}_i^t$ for each individual $i$ based on the keys corresponding to its selected individuals $\mathbf{X}_i^t$. Then, the mutant individual $\mathbf{v}_i^t$ is obtained by multiplying $\mathbf{s}_i^t$ with $\mathbf{X}_i^t$. Next, the LCrM Tokenizer processes $\mathbf{V}^t$ in the same manner as the LMutM Tokenizer to generate $\mathbf{F}^t$, and then feeds $[\mathbf{E}^t, \mathbf{F}^t]$ into the Differentiable LCrM. The Differentiable LCrM produces the candidate population $\mathbf{U}^t$ through two cross-attention mechanisms. The first mechanism determines, for each parent individual, the subset of mutant individuals from $\mathbf{V}^t$ it should attend to. The second mechanism, a deformable cross-attention module [50], computes the attention weights between each parent and the mutant individuals it attends to. The candidate individual is then obtained as a weighted combination of the parent and the attended mutants. Finally, $\mathbf{X}^t$ and $\mathbf{U}^t$ are passed to the Selection Module (SM) to generate the offspring population $\mathbf{X}^{t+1}$.

### 3.2.2 Tokenizer

**1) LMutM Tokenizer.** In POM, the dimensions of the inputs ($\mathbf{H}^t$ or $\mathbf{Z}^t$) to LMutM and LCrM are 2 and 3, respectively. However, such low-dimensional representations may fail to adequately capture the characteristics of the optimization landscape, limiting the information available to LMutM and LCrM. To address issue 3) in Section 3.1.2, we design a tokenizer that maps each individual in the population from the problem dimension $d$ to a fixed dimension $\hat{d}_t$ using a cross-attention mechanism:

$$
\begin{aligned}
\hat{\mathbf{x}}_i^t &= \begin{cases} \mathbf{x}_i^t, & \text{if } d \text{ is even} \\ append(\mathbf{x}_i^t, 0), & \text{otherwise} \end{cases} \\
\hat{\mathbf{K}}_T &= reshape(\hat{\mathbf{x}}_i^t, (-1, 2)) \\
\mathbf{K}_T &= \hat{\mathbf{K}}_T \times \mathbf{W}_{1K} + \mathbf{b}_{1K}, \quad \mathbf{V}_T = \hat{\mathbf{K}}_T \times \mathbf{W}_{1V} + \mathbf{b}_{1V} \\
\hat{\mathbf{e}}_i^t &= Tanh\left(\frac{\mathbf{Q}_T \times \mathbf{K}_T^{\mathrm{T}}}{\sqrt{d_{1V}}}\right) \times \mathbf{V}_T, \quad \mathbf{e}_i^t = reshape(\hat{\mathbf{e}}_i^t, (-1,))
\end{aligned}
\tag{3}
$$

where $\mathbf{x}_i^t \in \mathbb{R}^d$ is the $i$-th individual in the $t$-th generation, The $append$ and $reshape$ operations are functionally equivalent to their PyTorch counterparts. $\mathbf{W}_{1K} \in \mathbb{R}^{2 \times d_{1QK}}$, $\mathbf{W}_{1V} \in \mathbb{R}^{2 \times d_{1V}}$, $\mathbf{b}_{1K}$ and $\mathbf{b}_{1V}$ are bias vectors with dimensions $d_{1QK}$ and $d_{1V}$, respectively. $\mathbf{Q}_T \in \mathbb{R}^{d_T \times d_{1QK}}$ represents the shared query across the population. All parameters, including $\mathbf{W}_{1K}$, $\mathbf{W}_{1V}$, $\mathbf{b}_{1K}$, $\mathbf{b}_{1V}$, and $\mathbf{Q}_T$, are learnable.

By using Eq.(3), we obtain a $d_T \times d_{1V}$-dimensional token for each individual $\mathbf{x}_i^t$, and subsequently construct the population token matrix $\mathbf{E}^t = [\mathbf{e}_1^t, \mathbf{e}_2^t, \cdots, \mathbf{e}_n^t]$. Then we concatenate $\mathbf{E}^t$ with the token matrix $\mathbf{H}^t$ from the LMutM of POM to form $\mathbf{G}^t = [\mathbf{H}^t | \mathbf{E}^t]$, where $\mathbf{G}^t \in \mathbb{R}^{n \times \hat{d}_t}$ and $\hat{d}_t = 2 + d_T \times d_{1V}$.

**2) LCrM Tokenizer.** Similarly to the LMutM Tokenizer, the LCrM Tokenizer transforms the mutant population $\mathbf{V}^t$ into tokens $\mathbf{F}^t$ with a fixed dimension $d_t$ according to Eq. (3), where $d_t = d_T \times d_{1V}$.

### 3.2.3 Efficient LMutM

For issues 1) and 2) in Section 3.1.2, we introduce Efficient LMutM, which leverages an deformable attention mechanism to allow individuals to adaptively select a subset of individuals for information exchange. This approach avoids global interactions, improving the model's performance and diversity. Additionally, we incorporate a dropout layer that is active during both the training and testing phases within the EPOM, introducing more randomness to the model.

We adapt two self-attention mechanism in the Efficient LMutM. The first self-attention identifies the individuals within the parent population that each individual should atttend to according to the input

tokens $\mathbf{G}^t$. The attended individuals can be computed using Eq. (4).

$$\mathbf{Q}_{1M} = \mathbf{G}^t \times \mathbf{W}_{2Q} + \mathbf{b}_{2Q}, \quad \mathbf{K}_{1M} = \mathbf{G}^t \times \mathbf{W}_{2K} + \mathbf{b}_{2K}$$

$$\mathbf{V}_{1M} = \mathbf{G}^t \times \mathbf{W}_{2V} + \mathbf{b}_{2V}, \quad \hat{\mathbf{N}}_t = Softmax(\frac{\mathbf{Q}_{1M} \times \mathbf{K}_{1M}^{\mathrm{T}}}{\sqrt{d_{2V}}}) \times \mathbf{V}_{1M}, \quad (4)$$

$$\mathbf{N}_t = top\text{-}p\text{-}sampling(\hat{\mathbf{N}}_t | p)$$

where $\mathbf{W}_{2Q}, \mathbf{W}_{2K} \in \mathbb{R}^{\hat{d}_t \times d_{2QK}}$, $\mathbf{b}_{2Q}, \mathbf{b}_{2K} \in \mathbb{R}^{d_{2QK}}$, $\mathbf{W}_{2V} \in \mathbb{R}^{\hat{d}_t \times d_{out}}$, and $\mathbf{b}_{2V} \in \mathbb{R}^{d_{out}}$ denote learnable parameters, and $d_{out}$ is the fixed output dimension. If the population size $n \leq d_{out}$, the first $n$ dimensions are used to represent the selection probability of each individual. Otherwise, selection is restricted to the first $d_{out}$ individuals. The $top\text{-}p\text{-}sampling$ function selects $top\text{-}m_i$ individuals via top-p sampling [51], ensuring $\sum_{i=1}^{m} prob(\mathbf{x}_i^t) = p$. Applying top-p sampling to $\hat{\mathbf{N}}_t$ yields the index matrix $\mathbf{N}_t \in \mathbb{R}^{n \times k}$, where $k = \max\{m_i\}_{i=1}^{n}$, representing the selected individuals for each member of the population.

The second mechanism, deformable self-attention [50], computes the attention weight vector $\mathbf{s}_i^t$ for each individual $i$. Then, we derive the mutant population $\mathbf{V}^t = [\mathbf{v}_1^t, \mathbf{v}_2^t, \cdots, \mathbf{v}_n^t]$, where $\mathbf{v}_i^t = \mathbf{s}_i^t \times \mathbf{X}_i^t$.

Table 1: Training Functions for EPOM and POM. $z_i = x_i - \omega_i$.

| ID | Functions | Range |
|---|---|---|
| TF1 | $\sum_i |x_i - \omega_i|$ | $x \in [-10, 10], \omega \in [-10, 10]$ |
| TF2 | $\sum_i |(x_i - \omega_i) + (x_{i+1} - \omega_{i+1})| + \sum_i |x_i - \omega_i|$ | $x \in [-10, 10], \omega \in [-10, 10]$ |
| TF3 | $\sum_i z_i^2$ | $x \in [-100, 100], \omega \in [-50, 50]$ |
| TF4 | $\max\{|z_i|, 1 \leq i \leq d\}$ | $x \in [-100, 100], \omega \in [-50, 50]$ |
| TF5 | $\sum_{i=1}^{d-1}(100(z_i^2 - z_{i+1})^2 + (z_i - 1)^2)$ | $x \in [-100, 100], \omega \in [-50, 50]$ |

### 3.2.4 Differentiable LCrM

To resolve the issue in LCrM, we propose a differentiable LCrM, replacing the gumbel-softmax trick with a weighted summation approach. Specifically, the differentiable LCrM takes $[\mathbf{E}^t, \mathbf{F}^t]$ as inputs and generates the crossover weight $\mathbf{cw}^t \in \mathbb{R}^{n \times 2}$. The $i$-th candidate individual can be computed by Eq. (5).

$$\mathbf{u}_i^t = cw_{i,0}^t \cdot \mathbf{x}_i^t + cw_{i,1}^t \cdot mean(\mathbf{V}_i^t), \quad (5)$$

where $\mathbf{V}_i^t \in \mathbb{R}^{j \times d}$ denotes $j$ mutant individuals selected by parent individual $\mathbf{x}_i^t$. The $mean$ function computes the average along each dimension of the input matrix. Through Eq. (5), we obtain the candidate population $\mathbf{U}^t = [\mathbf{u}_1^t, \mathbf{u}_2^t, \cdots, \mathbf{u}_n^t]$. The remaining task is handled by **SM**, which peforms a one-to-one selection strategy to generate the offspring population $\mathbf{X}^{t+1}$.

The training set, and training strategy of EPOM are consistent with those of POM, as detailed in the Appendix A. After EPOM has been trained, we use Algorithm 1 to solve new problems.

### 3.2.5 Loss Function

The loss function of POM for the $i$-th training function consists of two components: 1) the difference in the mean fitness of adjacent generations, normalized $l_i^1$; and 2) the mean standard deviation across the dimensions of the population $l_i^2$. Ideally, $l_i^1$ encourages POM to converge toward the optimal point, and $l_i^2$ encourages population diversity. However, in practical applications, we found that $l_i^2$ has a limited effect on population diversity, therefore, we replaced it with crowding distance [52]. The loss function of EPOM for the $i$-th training function can be discribed as Eq.(6).

---

**Algorithm 1** Driving EPOM to Solve Problem

---

**Input:** Generations $T$, population size $n$, BBO problem $f$.
**Output:** The optimal $\mathbf{X}^T$ found.
1: EPOM loads the trained parameter $\theta$.
2: Randomly sample an initial population $\mathbf{X}^0$ of size $n$.
3: **for** $t = 0, 1, \dots, T-1$ **do**
4:     Construct $\mathbf{E}^t$ based on $\mathbf{X}^t$ and $f$ using LMutM Tokenizer.
5:     $\mathbf{G}^t \leftarrow [\mathbf{H}^t | \mathbf{E}^t]$.
6:     $\mathbf{S}^t \leftarrow$ Efficient LMutM$(\mathbf{G}^t)$.
7:     $\mathbf{V}^t \leftarrow \mathbf{S}^t \times \mathbf{X}^t$.
8:     Build $\mathbf{F}^t$ based on $\mathbf{V}^t$ using LCrM Tokenizer.
9:     $\mathbf{U}^t \leftarrow$ Differentiable LCrM$(\mathbf{E}^t, \mathbf{F}^t)$.
10:     $\mathbf{X}^{t+1} \leftarrow$ SM$(\mathbf{X}^t, \mathbf{U}^t)$.
11: **end for**

---

$$
\begin{aligned}
l_i^t &= l_i^1 + \alpha l_{cd} \\
&= \frac{\frac{1}{|\mathbf{X}^t|} \sum_{\mathbf{x} \in \mathbf{X}^t} f_i(\mathbf{x}|\omega^i) - \frac{1}{|\mathbf{X}^{t-1}|} \sum_{\mathbf{x} \in \mathbf{X}^{t-1}} f_i(\mathbf{x}|\omega^i)}{\left| \frac{1}{|\mathbf{X}^{t-1}|} \sum_{\mathbf{x} \in \mathbf{X}^{t-1}} f_i(\mathbf{x}|\omega^i) \right|} + \frac{\alpha}{|\mathbf{X}^t|} \sum_{i=2}^{n-1} \frac{f_i(\mathbf{x}_{i+1:n}^t|\omega^i) - f_i(\mathbf{x}_{i-1:n}^t|\omega^i)}{f_i(\mathbf{x}_{n:n}^t|\omega^i) - f_i(\mathbf{x}_{1:n}^t|\omega^i)} \quad (6)
\end{aligned}
$$

where $\mathbf{x}_{i:n}^t$ denotes the individual ranked $i$-th in ascending order (for a minimization problem) in the $t$-th generation. $\alpha$ is a hyperparameter, and we found that setting it to 0.1 better balances the convergence of the population toward the optimal value and its diversity.

## 4 Experiments

### 4.1 Experimental Setup

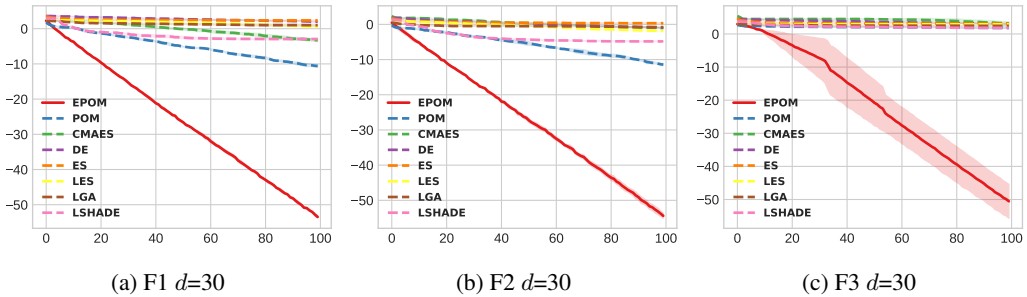

(a) F1 $d=30$        (b) F2 $d=30$        (c) F3 $d=30$

Figure 2: Part of the convergence curves of EPOM and other baselines. It shows the logarithmic convergence curve of these algorithms on functions in BBOB. See Appendix C for more details.

**Baselines.** Our core approach is a population-based pretraining BBO algorithm; therefore, we focus on comparisons with other population-based methods rather than non-population methods such as Bayesian optimization. Moreover, we do not compare with LLM-based approaches [30, 31, 32, 33, 34, 37, 38, 39], as these methods lack zero-shot optimization capabilities.

*Heuristic Population-based BBO Algorithms.* We compare against the following established population-based BBO algorithms: (a) **DE (DE/rand/1/bin)** [53]: A classic numerical optimization algorithm. (b) **ES ($\mu,\lambda$-ES)**: A well-known variant of evolutionary strategies. (c) **CMA-ES** [18]: Often considered the state-of-the-art method for continuous domain optimization in challenging settings (e.g., ill-conditioned, non-convex, multimodal problems). (d) **L-SHADE** [19]: A state-of-the-art variant of DE.

Table 2: Results of BBOB. EPOM is trained on TF1-TF5 with $d$ =10. The best results are indicated in bold, and the suboptimal results are underlined. The "+/=/-" at the bottom indicates that EPOM's performance is better/same than the given baseline on the given dimension setting and function.

| $d$ | F | EPOM | POM | ES | DE | CMA-ES | LSHADE | LES | LGA |
|---|---|---|---|---|---|---|---|---|---|
| 30 | F1 | **4.92E-54(5.26E-54)** | 3.72E-11(3.95E-11) | 2.30E+02(1.36E+01) | 9.46E+01(1.17E+01) | 7.79E-04(8.97E-04) | 1.28E-03(7.36E-04) | 4.93E+00(4.94E+00) | 1.13E+01(7.12E+00) |
| | F2 | **8.39E-55(8.27E-55)** | 4.69E-12(3.85E-12) | 2.18E+00(5.24E-01) | 1.10E-01(4.35E-03) | 8.45E-02(1.64E-02) | 1.47E-05(2.12E-06) | 1.45E-02(6.38E-03) | 1.75E-01(5.88E-02) |
| | F3 | **1.64E-44(2.84E-44)** | 6.57E+01(1.49E+01) | 1.41E+03(1.26E+02) | 1.02E+03(5.47E+01) | 2.47E+03(2.39E+03) | 7.12E+01(9.31E+00) | 8.10E+02(1.27E+02) | 2.82E+02(2.03E+01) |
| | F4 | **1.92E-40(3.31E-40)** | 6.95E+01(4.28E+01) | 3.35E+03(6.76E+02) | 1.99E+03(5.61E+02) | 2.21E+02(1.02E+00) | 1.04E+02(4.24E+00) | 6.11E+01(1.50E+02) | 3.76E+02(3.85E+01) |
| | F5 | 3.28E+02(5.05E+01) | 3.61E+01(3.13E+01) | 5.91E+01(4.64E+00) | 1.32E+00(2.70E-01) | **0.00E+00(0.00E+00)** | **0.00E+00(0.00E+00)** | 1.99E+02(5.46E+01) | **0.00E+00(0.00E+00)** |
| | F6 | **1.90E-47(2.90E-47)** | 1.69E-09(1.62E-09) | 3.97E+02(9.66E+00) | 5.29E+02(1.74E+02) | 8.99E-02(6.01E-03) | 1.54E-01(8.94E-02) | 1.11E+01(9.11E+00) | 2.25E+01(6.53E+00) |
| | F7 | **6.83E-37(7.89E-37)** | 3.78E-13(3.38E-13) | 1.61E+03(5.19E+01) | 7.62E+03(9.02E+02) | 3.44E+00(7.67E-01) | 1.25E+01(6.46E+00) | 1.20E+02(4.80E+01) | 6.97E+01(2.45E+01) |
| | F8 | **0.00E+00(0.00E+00)** | 6.23E-06(4.98E-06) | 4.21E+05(6.85E+04) | 3.26E+05(4.66E+04) | 3.15E+02(3.90E+02) | 3.08E+01(3.53E+00) | 3.01E+02(2.79E+03) | 1.63E+03(3.60E+02) |
| | F9 | 1.86E+02(8.10E-01) | 1.60E+02(1.94E+01) | 4.44E+05(8.88E+04) | 7.06E+05(9.64E+04) | **4.17E+01(1.20E+01)** | 5.85E+01(5.42E+01) | 2.37E+03(8.49E+02) | 1.38E+03(5.39E+02) |
| | F10 | **5.58E-48(9.24E-48)** | 2.24E+03(2.11E+03) | 3.56E+06(1.48E+06) | 2.33E+07(6.30E+06) | 3.39E+05(1.18E+05) | 1.16E+04(5.72E+03) | 7.58E+04(3.58E+04) | 2.67E+05(6.29E+04) |
| | F11 | **3.99E-48(4.42E-48)** | 7.38E+00(8.33E-01) | 1.59E+03(5.31E+02) | 5.73E+03(8.62E+02) | 5.55E+03(1.21E+03) | 1.53E+02(1.13E+02) | 2.36E+02(3.17E+01) | 3.95E+02(1.72E+02) |
| | F12 | **1.46E-46(1.19E-46)** | 5.13E-04(4.11E-04) | 4.18E+09(4.62E+08) | 1.37E+10(8.87E+08) | 2.91E+11(2.89E+10) | 4.10E+05(4.53E+05) | 1.04E+08(7.97E+07) | 9.59E+07(3.51E+07) |
| | F13 | **3.61E-25(2.49E-25)** | 6.76E-05(3.90E-05) | 1.57E+03(6.29E+01) | 1.07E+03(8.97E+01) | 9.66E+00(1.62E+00) | 2.44E+00(1.41E+00) | 8.61E+00(4.07E+01) | 2.40E+02(4.05E+01) |
| | F14 | **8.36E-29(7.43E-29)** | 2.29E-04(6.42E-05) | 9.04E+01(1.08E+01) | 5.84E+02(8.93E+01) | 1.92E+00(1.14E+00) | 4.38E-02(2.39E-02) | 6.01E+00(1.88E+00) | 4.02E+00(9.65E-01) |
| | F15 | **2.74E-03(4.75E-03)** | 7.84E+01(2.37E+01) | 1.62E+03(1.29E+02) | 4.31E+03(6.26E+02) | 4.27E+04(4.06E+04) | 1.16E+02(1.08E+01) | 8.73E+02(2.02E+02) | 2.84E+02(2.33E+01) |
| | F16 | 3.54E+01(5.25E+00) | 2.55E+01(1.42E+00) | 4.62E+01(4.62E+00) | 5.44E+01(5.74E+00) | 3.18E+01(3.66E+00) | 1.64E+01(5.59E+00) | **7.17E+00(1.16E+00)** | 3.24E+01(1.07E+00) |
| | F17 | **1.46E-27(5.82E-28)** | 2.79E-05(1.29E-05) | 2.47E+01(7.36E+00) | 2.43E+01(4.93E+00) | 3.78E-01(6.36E-02) | 4.67E-01(9.78E-02) | 9.74E+00(4.15E+00) | 2.21E+00(3.62E-01) |
| | F18 | **4.56E-27(3.83E-27)** | 1.30E-01(1.86E-01) | 9.84E+01(1.68E+01) | 1.19E+02(4.66E+00) | 2.26E+00(5.51E-01) | 9.34E-01(3.55E-01) | 3.43E+01(1.25E+01) | 1.21E+01(3.22E+00) |
| | F19 | **4.41E+00(6.06E-01)** | 4.82E+00(2.55E-01) | 5.43E+01(4.16E+00) | 5.00E+01(1.17E+01) | 5.94E+00(4.07E-01) | 5.44E+00(4.67E-01) | 1.61E+01(2.59E+00) | 7.06E+00(2.56E-01) |
| | F20 | -1.69E+00(4.86E+00) | -1.32E+01(3.32E+00) | 1.25E+05(5.14E+04) | 1.08E+05(2.56E+04) | 3.13E+00(9.10E-02) | 3.13E+00(1.03E+00) | **-2.72E+01(1.26E+01)** | 9.09E+01(1.05E+02) |
| | F21 | 7.63E+01(2.03E+00) | 3.36E+01(1.03E+01) | 8.80E+01(6.16E-01) | 8.56E+01(7.64E-01) | **2.89E+00(5.34E-02)** | 1.44E+01(1.26E+01) | 1.99E+01(9.86E+00) | 9.98E+00(2.89E+00) |
| | F22 | 7.35E+01(1.47E+00) | 1.57E+01(1.03E+01) | 8.92E+01(1.82E+00) | 8.57E+01(6.39E-01) | 1.96E+00(5.02E-03) | **1.14E+00(7.17E-01)** | 1.68E+01(4.44E+00) | 9.91E+00(5.50E+00) |
| | F23 | 3.26E+00(1.70E-01) | 3.68E+00(2.22E-01) | 3.61E+00(5.29E-01) | 3.49E+00(3.33E-01) | 3.75E+00(6.23E-01) | 3.38E+00(3.16E-01) | **3.01E+00(4.00E-01)** | 4.38E+00(1.38E-01) |
| | F24 | 3.33E+02(1.22E+01) | 2.81E+02(2.11E+01) | 2.82E+02(2.17E+01) | 3.20E+02(3.56E+01) | 2.08E+02(2.20E+01) | **1.84E+02(3.16E+01)** | 7.08E+02(7.80E+01) | 3.69E+02(4.32E+01) |
| 100 | F1 | **2.33E-53(2.03E-53)** | 1.48E-11(8.37E-12) | 1.60E+03(3.45E+01) | 4.62E+03(5.31E+02) | 4.34E+01(4.29E+00) | 1.64E+01(8.78E-01) | 2.20E+02(4.99E+01) | 1.13E+02(2.07E+01) |
| | F2 | **4.26E-55(5.23E-55)** | 4.70E-12(2.55E-12) | 4.08E+01(6.19E+00) | 2.24E+01(3.00E+00) | 4.17E+01(9.20E+00) | 7.58E-02(4.38E-02) | 4.56E+00(1.29E+00) | 3.28E+00(5.75E-01) |
| | F3 | 3.36E-01(5.82E-01) | **1.07E-09(1.16E-09)** | 1.06E+04(7.18E+02) | 4.77E+04(2.87E+03) | 3.24E+04(8.39E+03) | 8.71E+02(9.08E+01) | 2.72E+03(1.89E+02) | 1.82E+03(5.85E+01) |
| | F4 | 5.02E+03(8.69E+03) | **1.39E-07(1.93E-07)** | 6.28E+04(7.31E+03) | 2.96E+05(2.45E+04) | 3.77E+03(3.11E+02) | 1.29E+03(1.69E+02) | 5.15E+03(1.25E+03) | 2.49E+03(1.51E+02) |
| | F5 | 1.82E+03(4.32E+01) | 3.04E+02(4.14E+01) | 2.03E+01(1.42E+01) | 4.64E+00(2.13E+00) | 1.63E+02(2.83E+02) | **3.98E+00(4.69E+00)** | 1.30E+01(1.07E+00) | 4.05E+00(4.50E+00) |
| | F6 | **4.82E-46(8.34E-46)** | 9.32E-10(5.17E-10) | 2.82E+04(2.04E+02) | 9.15E+03(2.22E+02) | 2.65E+02(1.05E+02) | 4.00E+01(5.60E+00) | 4.37E+02(4.98E+01) | 2.09E+02(1.11E+01) |
| | F7 | **9.29E-30(1.61E-29)** | 2.42E-13(1.55E-13) | 1.11E+04(2.21E+03) | 6.11E+04(4.18E+03) | 2.79E+03(3.55E+02) | 1.96E+02(7.71E+01) | 1.43E+03(3.93E+02) | 9.43E+02(3.33E+02) |
| | F8 | **0.00E+00(0.00E+00)** | 3.01E-08(1.57E-08) | 2.09E+07(2.75E+05) | 1.60E+08(2.16E+07) | 6.06E+04(2.47E+04) | 1.35E+04(7.05E+03) | 2.40E+05(1.44E+04) | 9.43E+04(6.50E+04) |
| | F9 | 6.43E+02(9.70E-02) | **6.41E+02(8.48E-01)** | 1.97E+07(1.59E+06) | 2.18E+08(2.65E+07) | 1.12E+05(8.38E+04) | 4.06E+03(9.38E+02) | 3.71E+05(1.73E+04) | 1.07E+05(3.15E+04) |
| | F10 | **1.07E-20(1.85E-20)** | 2.34E+01(4.04E+01) | 5.73E+07(1.15E+07) | 3.29E+08(1.13E+07) | 7.27E+07(4.91E+07) | 4.19E+05(3.99E+04) | 2.82E+06(1.24E+06) | 3.83E+06(6.94E+05) |
| | F11 | **9.46E-03(1.64E-02)** | 1.71E+01(1.48E+00) | 4.63E+03(5.42E+02) | 2.41E+04(2.95E+02) | 3.25E+04(5.30E+03) | 4.59E+02(8.92E+01) | 7.82E+02(4.67E+01) | 9.26E+02(1.23E+02) |
| | F12 | **4.95E-47(1.95E-47)** | 1.43E-04(1.06E-04) | 4.15E+10(1.75E+09) | 4.86E+11(8.89E+10) | 1.91E+12(5.61E+11) | 9.01E+08(4.73E+08) | 3.83E+09(3.59E+08) | 2.12E+09(1.32E+09) |
| | F13 | **1.52E-24(2.32E-24)** | 7.23E-05(7.37E-05) | 4.18E+03(8.22E+01) | 6.65E+03(4.61E+02) | 6.35E+02(1.15E+02) | 3.89E+02(6.17E+01) | 1.53E+03(1.26E+02) | 9.26E+02(7.82E+01) |
| | F14 | **1.49E-27(1.98E-27)** | 9.07E-05(6.57E-05) | 4.51E+02(6.12E+01) | 3.85E+03(5.14E+02) | 4.15E+02(6.83E+01) | 7.45E+00(3.08E+00) | 3.57E+01(6.65E+00) | 4.11E+01(5.20E+00) |
| | F15 | **6.06E-47(1.05E-46)** | 4.88E+02(4.17E+01) | 9.88E+02(7.02E+02) | 6.86E+04(8.80E+03) | 3.37E+04(1.93E+04) | 1.05E+03(9.66E+01) | 3.61E+03(2.92E+02) | 1.65E+03(1.35E+01) |
| | F16 | 4.92E+01(1.43E+00) | 4.72E+01(4.72E-01) | 8.41E+01(3.87E+00) | 1.90E+02(1.40E+01) | 5.36E+01(3.48E+00) | 3.44E+01(3.21E+00) | **1.29E+01(6.39E-01)** | 5.58E+01(1.67E+00) |
| | F17 | **1.03E-26(1.74E-26)** | 5.50E-07(2.32E-07) | 1.26E+03(3.78E+02) | 1.77E+04(8.83E+02) | 5.71E+00(1.24E+00) | 2.61E+00(9.12E-02) | 2.10E+01(2.67E+00) | 1.20E+01(1.33E+00) |
| | F18 | **6.80E-26(1.10E-25)** | 5.94E-06(4.45E-06) | 1.73E+03(1.13E+02) | 2.66E+04(3.33E+03) | 2.65E+01(4.37E+00) | 1.06E+01(1.25E+00) | 5.66E+01(1.16E+01) | 4.16E+01(5.48E+00) |
| | F19 | **1.62E+00(1.29E+00)** | 6.74E+00(4.70E-01) | 5.37E+01(5.46E+01) | 5.32E+02(3.05E+03) | 1.08E+01(1.71E+00) | 8.95E+00(2.98E-01) | 2.75E+01(3.36E+00) | 1.30E+01(1.38E+00) |
| | F20 | -1.63E+00(6.36E+00) | **-5.08E+00(7.15E-01)** | 1.56E+06(8.58E+04) | 5.16E+06(8.26E+05) | 3.98E+04(1.36E+04) | 1.70E+03(3.56E+02) | 4.66E+04(2.02E+04) | 2.56E+04(7.96E+03) |
| | F21 | 7.75E+01(1.06E+00) | 4.03E+01(1.77E+00) | 2.10E+02(4.08E-01) | 1.22E+03(2.28E+02) | 6.56E+01(1.25E-01) | **1.37E+01(1.03E+00)** | 7.62E+01(7.83E-01) | 7.35E+01(9.27E-01) |
| | F22 | 8.28E+01(1.01E-01) | 5.95E+01(1.27E+00) | 2.33E+02(1.93E+01) | 1.46E+02(4.77E+00) | 7.02E+01(2.84E+00) | **3.12E+01(1.82E+01)** | 7.62E+01(5.88E+00) | 8.01E+01(7.78E+00) |
| | F23 | 4.35E+00(9.52E-01) | 4.83E+00(2.54E-01) | 5.14E+00(2.01E-01) | 4.84E+00(5.12E-01) | 5.23E+00(3.62E-01) | 5.12E+00(3.16E-01) | 5.21E+00(1.89E-01) | 7.15E+00(1.68E-01) |
| | F24 | 1.37E+03(3.12E+01) | 1.24E+03(1.23E+02) | 1.25E+03(5.46E+01) | 1.82E+03(1.75E+01) | 1.61E+03(6.24E+01) | **9.20E+02(5.85E+01)** | 3.52E+03(3.79E+02) | 3.37E+03(2.95E+02) |
| +/=/- | - | -/-/- | 32/0/16 | 44/0/4 | 45/0/3 | 38/0/10 | 36/0/12 | 38/0/10 | 40/0/8 |

These algorithms are implemented as follows: DE and ES using Geatpy [54], CMA-ES and IPOP-CMA-ES using cmaes[3], and L-SHADE using pyade[4].

*Pretrained BBO Algorithms.* For comparison with EPOM, we select three recent meta-learned BBO algorithms: (a) **POM** [13]: The Latest pretrained optimization model with the best performance. (b) **LES** [11]: A learnable evolutionary strategy that uses a data-driven approach to enhance generalization performance and search efficiency. (c) **LGA** [12]: A data-driven learnable genetic algorithm that adapts to unseen optimization problems, search dimensions, and evaluation budgets.

For all algorithms, we train EPOM and POM on the $TS$ problem, with a maximum of 100 evolution generations, $n = 100$, and a problem dimension of 10.

**Benchmarks** BBOB [55, 56] is a well-established benchmark suite for evaluating optimization algorithms. It includes a diverse set of high-dimensional continuous functions, encompassing single-peak, multi-peak, rotated, and distorted functions, along with functions exhibiting characteristics such as Lipschitz continuity and second-order differentiability.

### 4.2 Results

We evaluate the generalization ability of EPOM on 24 BBOB benchmark functions with dimensions $d = 30$ and $d = 100$. Table 2 summarizes the performance comparison across all algorithms (see

---

[3]https://github.com/CyberAgentAILab
[4]https://github.com/xKuZz/pyade

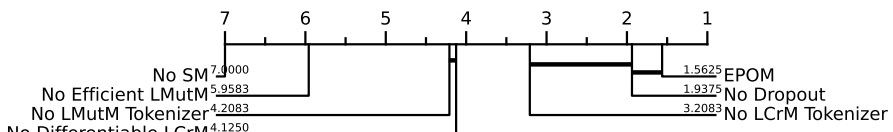

Figure 4: Results of ablation study on BBOB. The metric used to evaluate performance is the optimal value of the function found, with smaller values being better. Here, $d = 30$.

Appendix Figures 6 and 8 for detailed results). The fitness value of the best individual in the final generation for each function and dimension setting is reported.

Compared to POM, EPOM achieves superior results on 32 out of 48 functions across dimensions $d = 30$ and $d = 100$ (66.7%), indicating consistent performance gains across both low- and high-dimensional settings. This demonstrates EPOM's improved scalability and robustness. Moreover, EPOM outperforms all baseline methods in most cases, highlighting its strong generalization capability across diverse optimization landscapes.

EPOM exhibits significantly better convergence properties, as evidenced by its ability to achieve extremely small objective values. In contrast, POM often struggles to achieve comparable precision, particularly in high-dimensional problems. This highlights EPOM's enhanced ability to model population information and exploit fitness landscapes effectively.

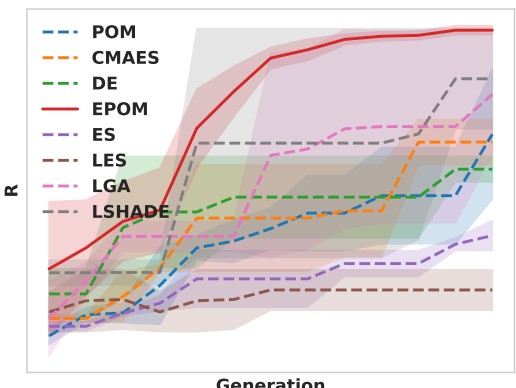

Figure 3: The results of robot control in the Bipedal Walker task.

**Bipedal Walker [57]**. The Bipedal Walker task aims to optimize a fully connected neural network with $d = 874$ parameters over $k = 800$ time steps to enhance robot locomotion control. In Fig. 3, EPOM achieves rapid and high-quality convergence, whereas LSHADE exhibits lower effectiveness, and CMA-ES, DE, and LES suffer from premature convergence.

## 4.3 Ablation Study

The results of ablation study for the designed modules are shown as Fig. 4. Configurations include the following items: (a) *No Efficient LMutM*: where the Efficient LMutM is excluded, and a simple *DE/rand/1/bin* mutation operator is employed; (b) *No LMutM Tokenizer*: the LMutM Tokenizer is excluded; (c) *No Diffrentiable LCrM*: indicating the absence of the learnable crossover operation, using only binomial crossover; (d) *No LCrM Tokenizer*: the LCrM Tokenizer is excluded; (e) *No Dropout*: the dropout layer for $\mathbf{S}^t$ is excluded; (f) *No SM*: the 1-to-1 selection module is excluded.

Upon observing the experimental results, we discovered that the performance of the model without SM deteriorated significantly. This is because the model loses its evolutionary direction. The impact of the modules can be roughly ranked as follows: *Efficient LMutM $\succ$ LMutM Tokenizer $\succ$ Differentiable LCrM $\succ$ LCrM Tokenizer $\succ$ Dropout*. This indicates that the Efficient LMutM introduces a more powerful representational ability to the model, addresses the problem of strategy inefficiency in POMs, and enhances the model's search capabilities. The LMutM Tokenizer can effectively extract the distribution characteristics of the population. The introduction of this characteristic significantly improves the model's decision-making ability, enabling it to better understand the fitness landscape. The Differentiable LCrM and LCrM Tokenizer can achieve information interaction between new and old individuals, strike an adaptive balance between exploration and exploitation, and enhance the generalization and convergence of the model.

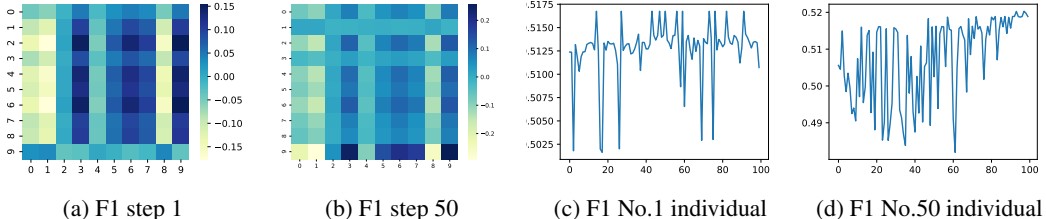

|                |                |                    |                     |
|:--------------:|:--------------:|:------------------:|:-------------------:|
| (a) F1 step 1  | (b) F1 step 50 | (c) F1 No.1 individual | (d) F1 No.50 individual |

Figure 5: (a)(b) Visualization results of some information selection strategies of Efficient LMutM. The numbers 0-9 represent the population ranking, and the smaller the ranking, the higher the fitness. The population size here is 10 and parameter $p$ for top-p-sampling is set to 1. (c)(d) Visualization results of crossover weight of some individuals of Differentiable LCrM. The x label represents the generation, and the y label denotes the crossover weight of $\mathbf{X}^t$. The population size is 100. (a), (b), (c) and (d) are all tested on BBOB $d = 30$

### 4.4 Visualization Analysis

**Analysis of Efficient LMutM** To conduct an in-depth analysis of the Efficient LMutM strategies, we visualize the evolution of $\mathbf{S}^t$ through heat maps, as shown in Fig. 5 (further details can be found in Appendix D). These heat maps reveal two key insights: 1) Similar to the original version, Efficient LMutM effectively balances exploration and exploitation by dynamically adjusting the weights of individuals based on their performance; and 2) Efficient LMutM adaptively generates diverse mutant strategies in response to varying task landscapes and individual performance.

**Analysis of Differentiable LCrM** We visualize the crossover strategies of Differentiable LCrM. As shown in Fig. 5 (refer to Appendix Figures 15-19 for additional details), Differentiable LCrM demonstrates its dynamic crossover strategies, generating variable crossover weights based on the overall performance of individuals and the characteristics of task landscapes.

## 5 Conclusion

We propose the Enhanced Pretrained Optimization Models, a novel framework designed to enhance the performance and robustness of POMs. The experimental results demonstrate that EPOM addresses the key limitations of POM, including insufficient task feature modeling, inefficient strategy generation, and unstable training process. By introducing the LMutM/LCrM Tokenizer, Efficient LMutM, and Differentiable LCrM, EPOM achieves state-of-the-art performance in zero-shot optimization, particularly in high-dimensional landscapes. These advancements make EPOM a highly promising candidate for real-world black-box optimization problems, offering superior scalability, robustness, and efficiency compared to existing methods.

EPOM's ability to outperform POM and other baselines across a wide range of functions, especially in high-difficulty scenarios, underscores its potential as a universal black-box optimizer. This work not only advances the state of the art in zero-shot optimization but also paves the way for broader adoption in machine learning and engineering applications.

However, the performance of EPOM in heterogeneous search spaces, involving optimization tasks with diverse data types (e.g., images, strings), remains underexplored. We identify this as an important avenue for future work.

## Acknowledgement

This work was supported in part by Natural Science Basic Research Program of Shaanxi under Grant 2025JC-QYCX-060, in part by the National Natural Science Foundation of China under Grant 62206205, 62472345, and 62471371, in part by the Young Talent Fund of Association for Science and Technology in Shaanxi, China under Grant 20230129, in part by the Guangdong High-level Innovation Research Institution Project under Grant 2021B0909050008, and in part by the Guangzhou Key Research and Development Program under Grant 202206030003.

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

# A  Tasks, Loss Function & MetaGBT

**Tasks**. We form a training task set $TS = \{f_i(\mathbf{X}|\omega^j)\}$, where $i \in [1,5]$ and $j \in [1,N]$, comprising $4N$ tasks derived from Table 1 in appendix, where $\omega_i$ denotes the task parameter influencing the function's landscape offset. Our selection of these functions for the training task is motivated by their diverse landscape features.

**MetaGBT**. The pseudocode for *MetaGBT* is presented in Algorithm 2. Initially, we sample the *EPOM* parameter $\theta$ from a standard normal distribution. The objective of *MetaGBT* is to iteratively update $\theta$ to bring it closer to the global optimum $\theta^*$. In line 2, we sample a population for each task in *TS*. Lines 3, 4 and 5 involve the resampling of task parameters for all tasks in *TS*, thereby altering the task landscape, augmenting training complexity, and enhancing the learning of robust optimization strategies by EPOM. The final loss function (line 10) is determined by computing the average of the loss functions for all tasks. Subsequently, in line 12, we update $\theta$ using a gradient-based optimizer, such as Adam [58]. The trained *EPOM* is then ready for application in solving an unknown BBO problem, as depicted in Algorithm 1.

---

**Algorithm 2** MetaGBT

---

**Input:** $T$, $n$, training set $TS$.
**Output:** The optimal $\theta$.
 1: Randomly sample the parameter $\theta$ of *EPOM*.
 2: **while** not done **do**
 3:     Sample $|TS|$ populations of size $n$ to obtain the population set $pop \leftarrow [\mathbf{X}_1^0, \mathbf{X}_2^0, \cdots, \mathbf{X}_{|TS|}^0]$.
 4:     **for** $i = 1, 2, \ldots, |TS|$ **do**
 5:       Randomly sample $\omega^i$ for the $f_i$ in $TS$.
 6:     **end for**
 7:     **for** $t = 1, 2, \ldots, T$ **do**
 8:       **for** $i = 1, 2, \ldots, |TS|$ **do**
 9:         $\mathbf{X}_i^t \leftarrow EPOM(\mathbf{X}_i^{t-1}, 1|\theta)$.
10:         $loss_i^t \leftarrow l_i(\mathbf{X}_i^t, \mathbf{X}_i^{t-1}, f_i, \omega^i, \lambda)$.
11:       **end for**
12:       $\theta \leftarrow$ Update $\theta$ using Adam based on $\frac{1}{|TS|}\sum_i loss_i^t$.
13:     **end for**
14: **end while**

---

# B Parameters

The primary control parameters of CMA-ES and L-SHADE are automatically adjusted. For LGA and LES, we utilized the optimal parameters provided by the authors without modifications. Other hyperparameters were tuned using grid search to identify the optimal combinations, and multiple experiments were conducted accordingly. Detailed parameter settings are presented in Table 3. Each experiment reports the mean and standard deviation of the results from various sets of experiments, with a consistent population size of 100 across all trials. All experiments are performed on a device with GeForce RTX 3090 24G GPU, Intel Xeon Gold 6126 CPU and 64G RAM.

Table 3: Detailed parameter settings for all baselines.

| Algorithm | item | setting |
|---|---|---|
| POM | $d_m = 1000$ $d_c = 4$ | Standard Settings for POM (M). |
| CMA-ES | Initial $\sigma = \frac{upper\_bounds+lower\_bounds}{2} * \frac{2}{5}$ | $2/5$ is a hyperparameter, and we determine this hyperparameter between $[0.1, 1]$ using a grid search, with a step of 0.1. |
| | Initial $\mu$ | $\mu = \textbf{lower\_bounds} + (randn(d) * (\textbf{upper\_bounds} - \textbf{lower\_bounds}))$, where $randn(d)$ stands for sampling a $d$-dimensional vector from a standard normal distribution. |
| LSHADE | $memory\_size = 6$ | We use a grid search to determine this parameter, the search interval is $[1, 10]$, and the search step is 1. |
| ES | $selFuc = urs$ | We use a grid search to determine this parameter, the search interval is $[dup, ecs, etour, otos, rcs, rps, rws, sus, tour, urs]$ [54]. |
| | $Nsel = 0.5$ | we determine this hyperparameter between $[0.1, 0.8]$ using a grid search, with a step of 0.1. |
| DE | $F = 0.5$ | we determine this hyperparameter between $[0.1, 0.9]$ using a grid search, with a step of 0.1. [54]. |
| | $XOVR = 0.5$ | we determine this hyperparameter between $[0.1, 0.9]$ using a grid search, with a step of 0.1. |
| LGA LES | All parameters | We use the pre-trained optimal parameters provided by the authors. |

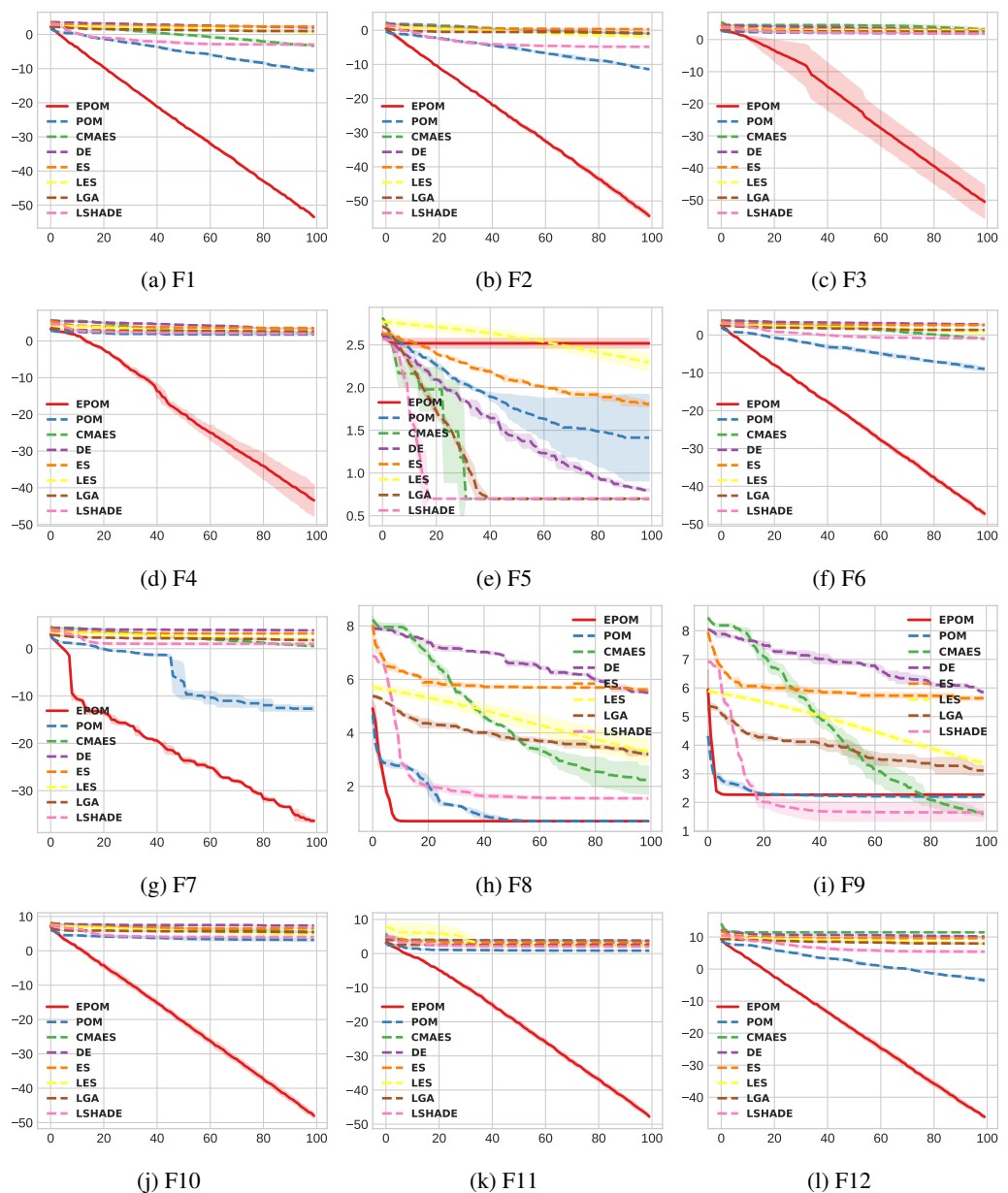

Figure 6: The log convergence curves of EPOM and other baselines on F1-F12. It shows the convergence curve of these algorithms on functions in BBOB with $d = 30$.

## C  Visualization Results of BBOB

The visualization results of all BBOB experiments are shown in Figure 6-9.

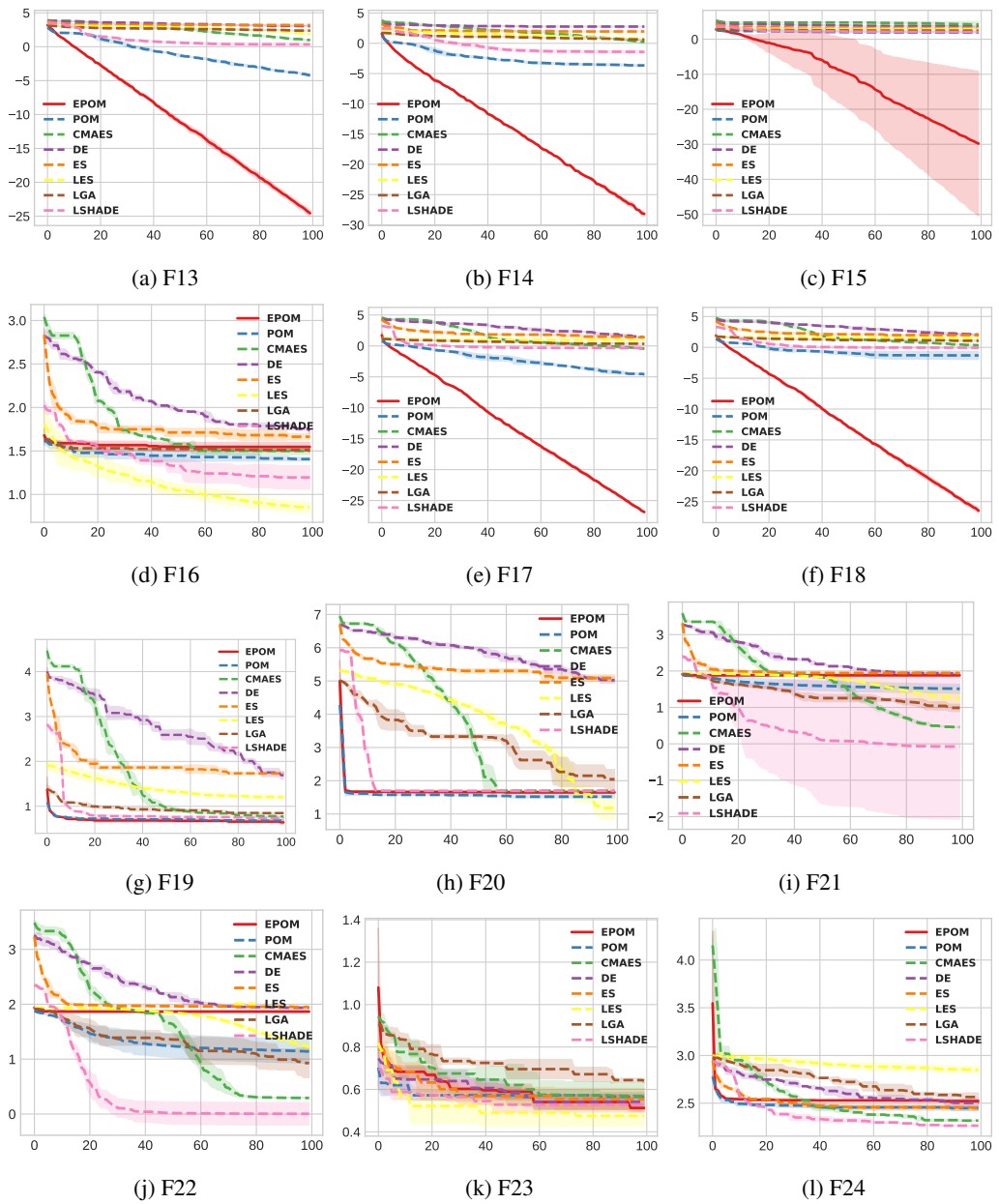

Figure 7: The log convergence curves of EPOM and other baselines on F13-F24. It shows the convergence curve of these algorithms on functions in BBOB with $d = 30$.

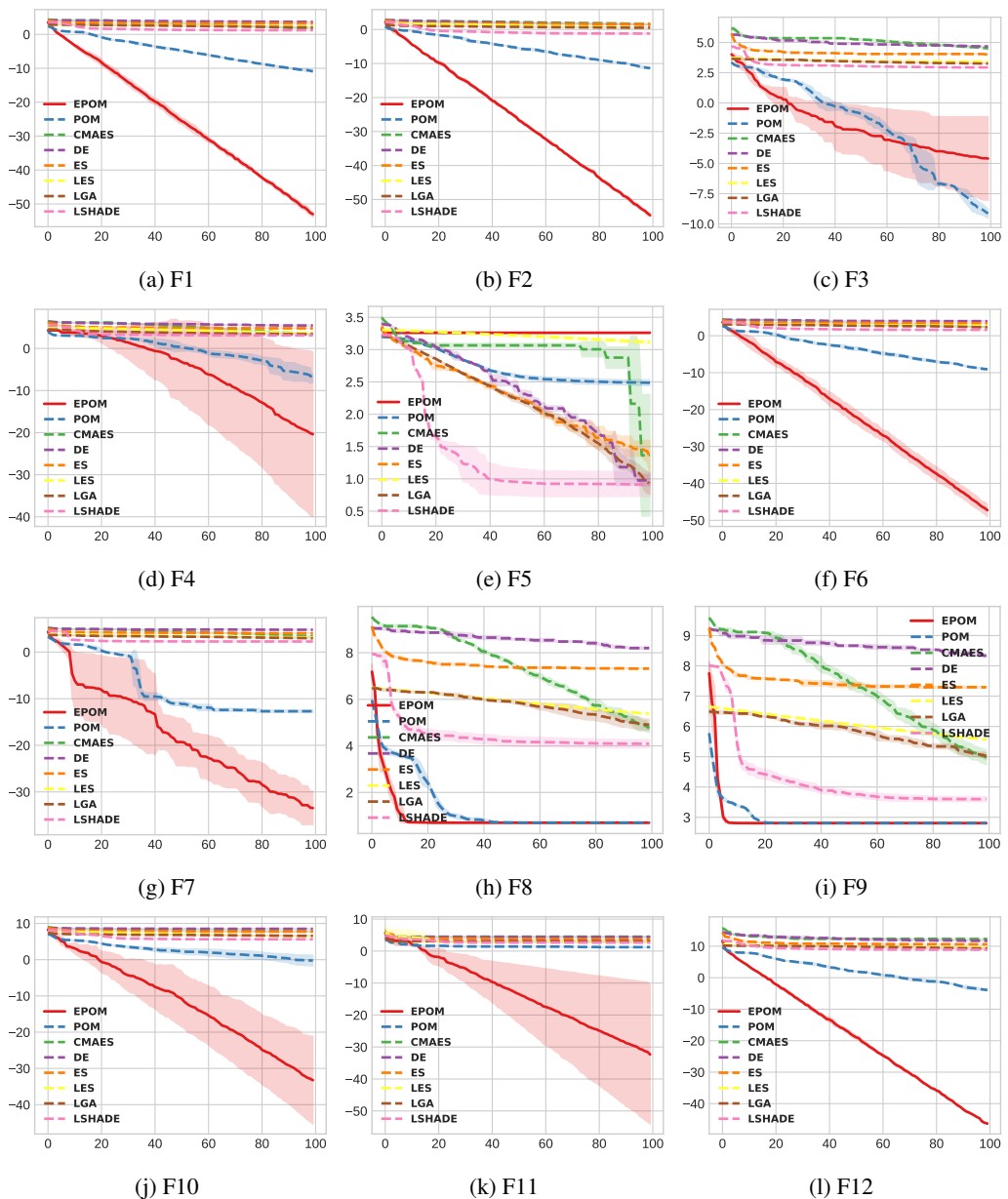

Figure 8: The log convergence curves of EPOM and other baselines on F1-F12. It shows the convergence curve of these algorithms on the functions in BBOB with $d = 100$.

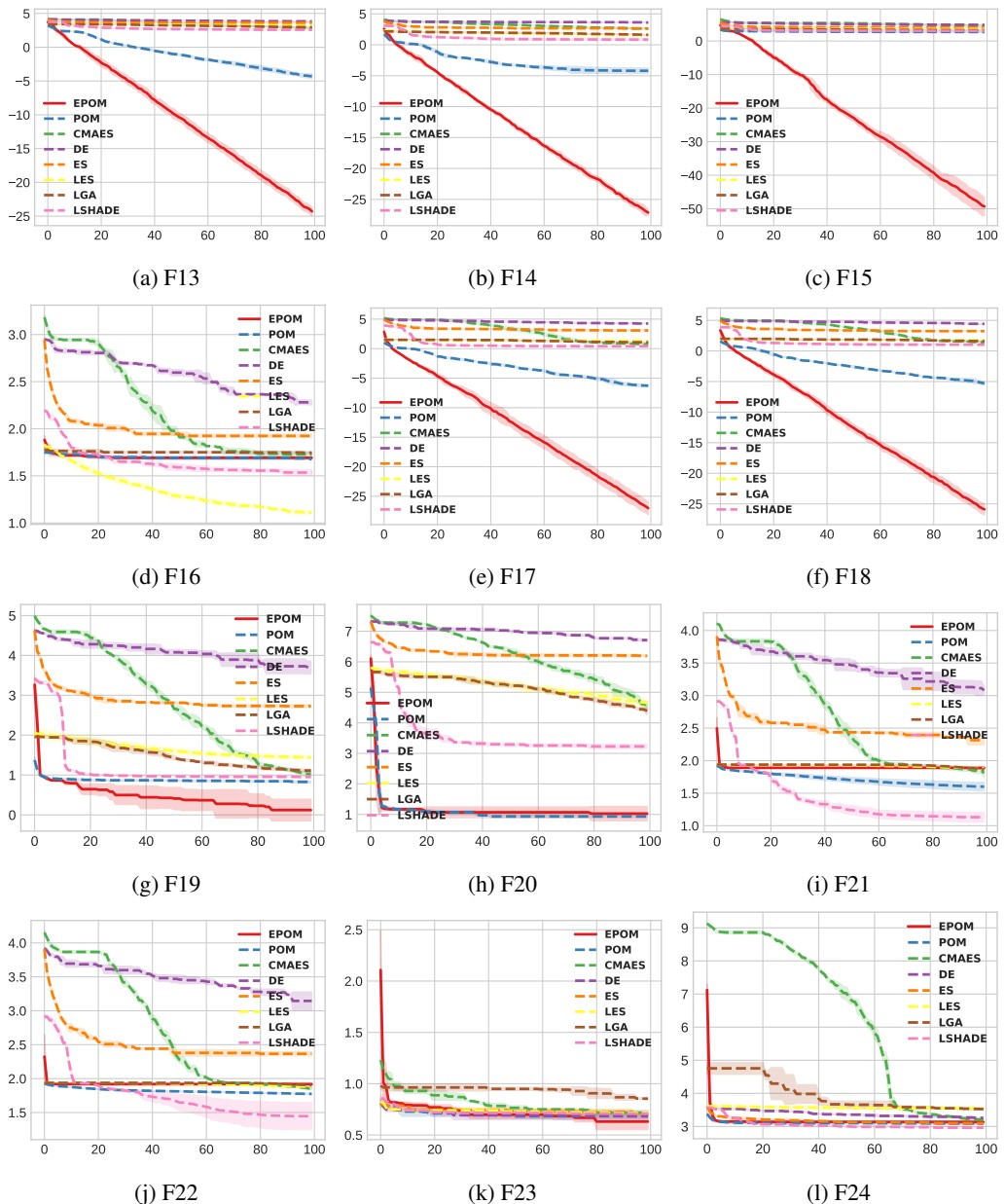

Figure 9: The log convergence curves of EPOM and other baselines on F13-F24. It shows the convergence curve of these algorithms on the functions in BBOB with $d = 100$.

# D  Visualization Results of Efficient LMutM

The visualization results of Efficient LMutM are shown in Figure 10-14.

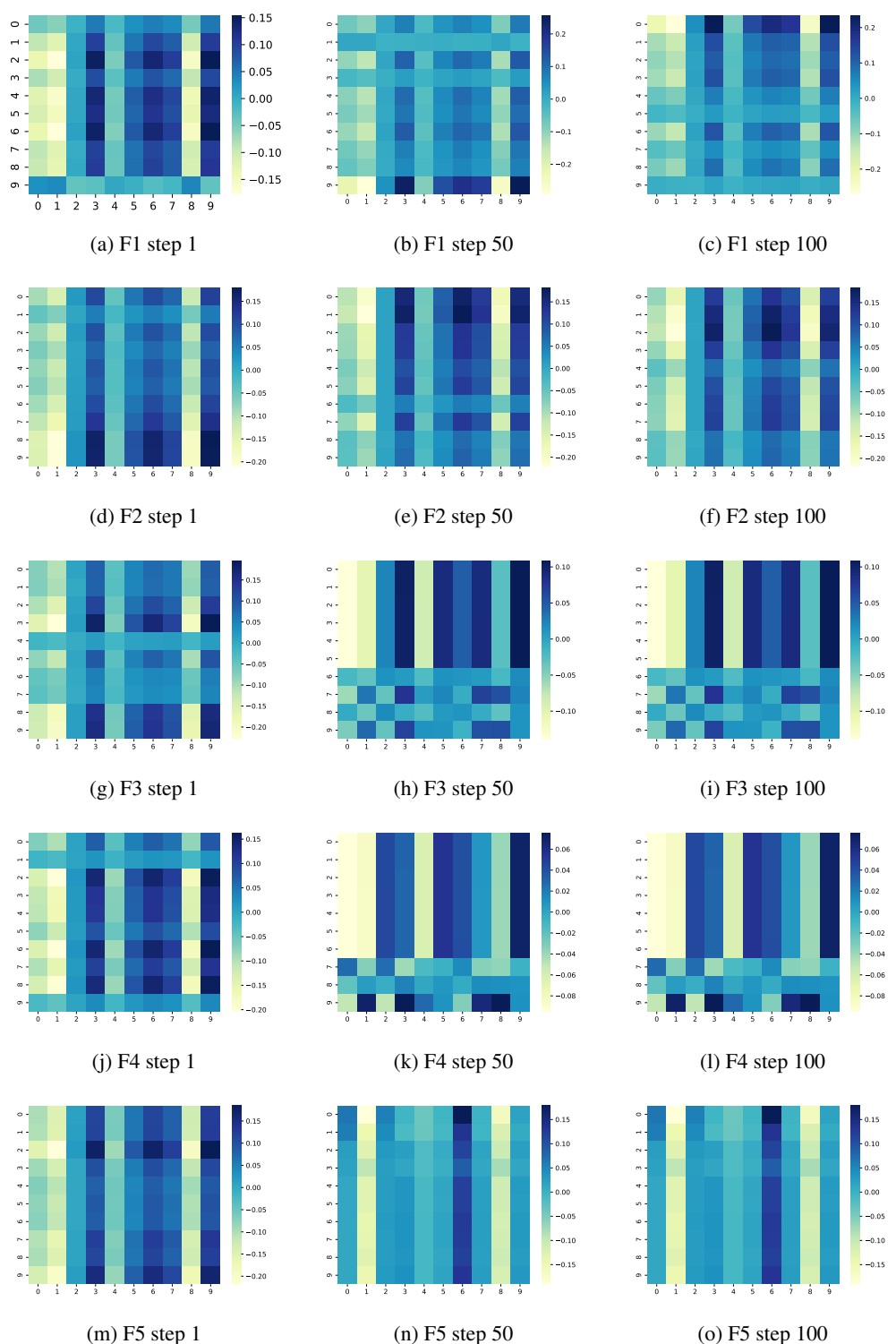

Figure 10: F1-F5 Visualization results of some information selection strategies of Efficient LMutM. 0->9 represents the population ranking, and the smaller the ranking, the higher the fitness.

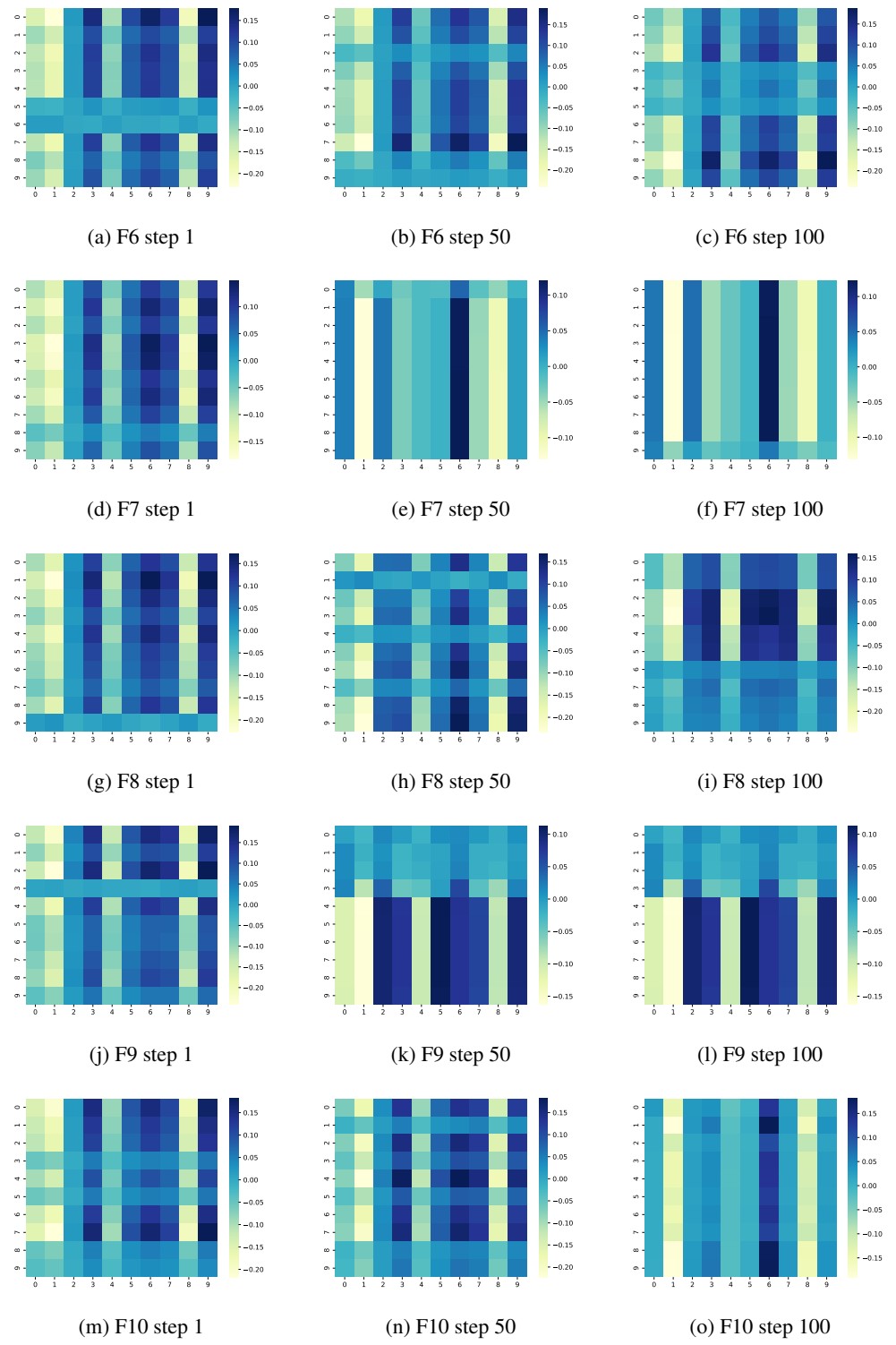

Figure 11: F6-F10 Visualization results of some information selection strategies of Efficient LMutM. 0->9 represents the population ranking, and the smaller the ranking, the higher the fitness.

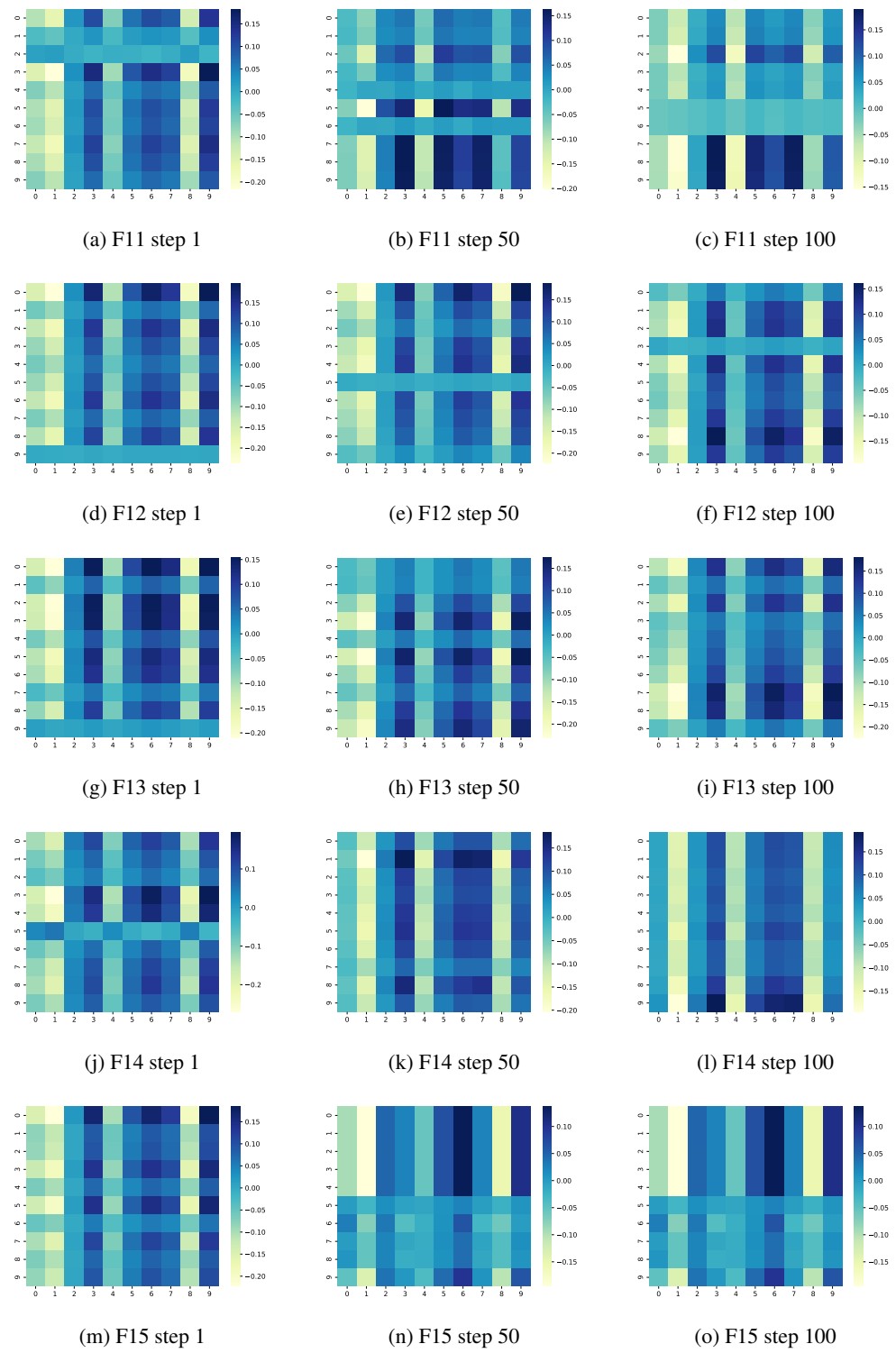

Figure 12: F11-F15 Visualization results of some information selection strategies of Efficient LMutM. 0->9 represents the population ranking, and the smaller the ranking, the higher the fitness.

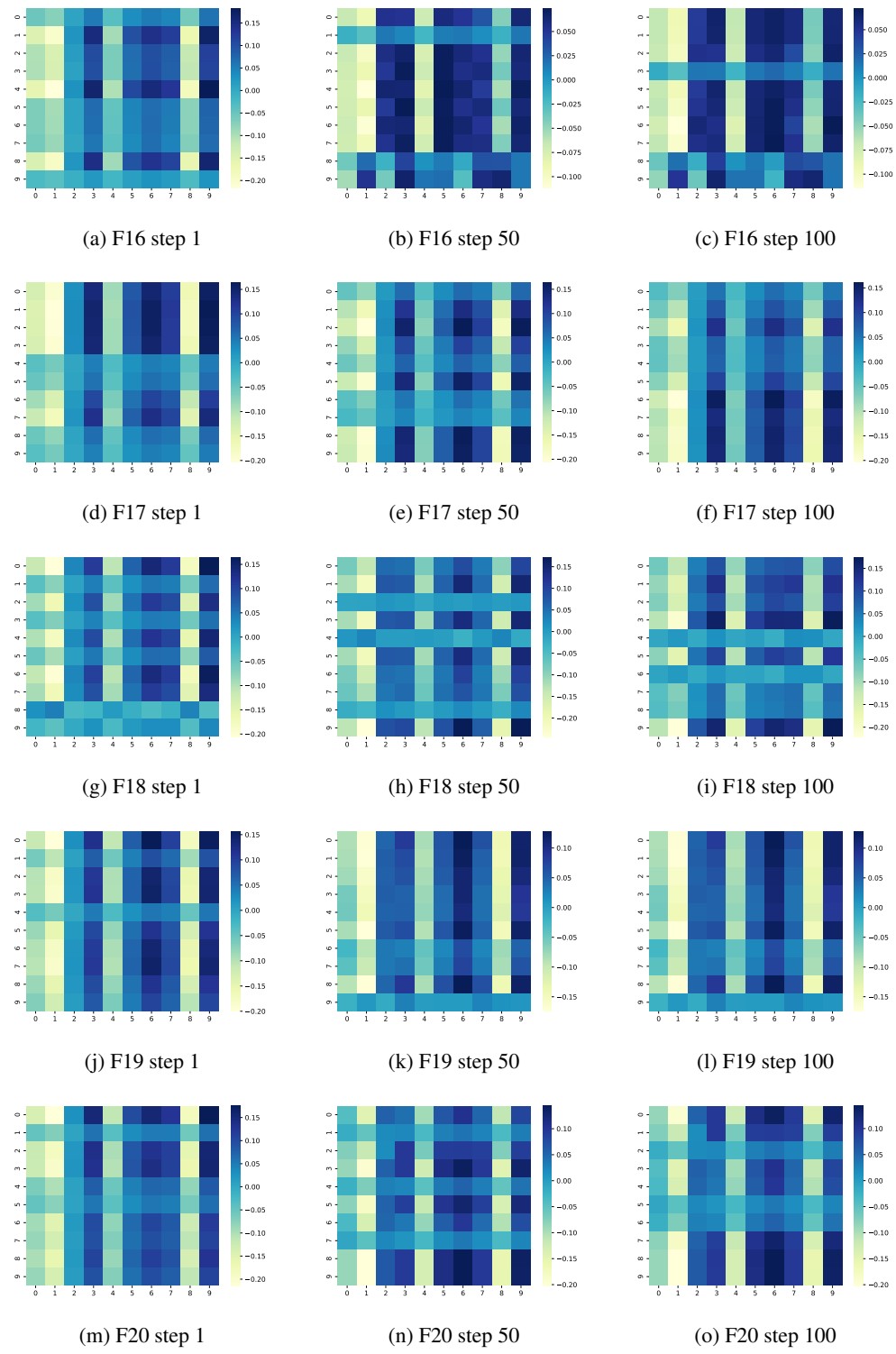

Figure 13: F16-F20 Visualization results of some information selection strategies of Efficient LMutM. 0->9 represents the population ranking, and the smaller the ranking, the higher the fitness.

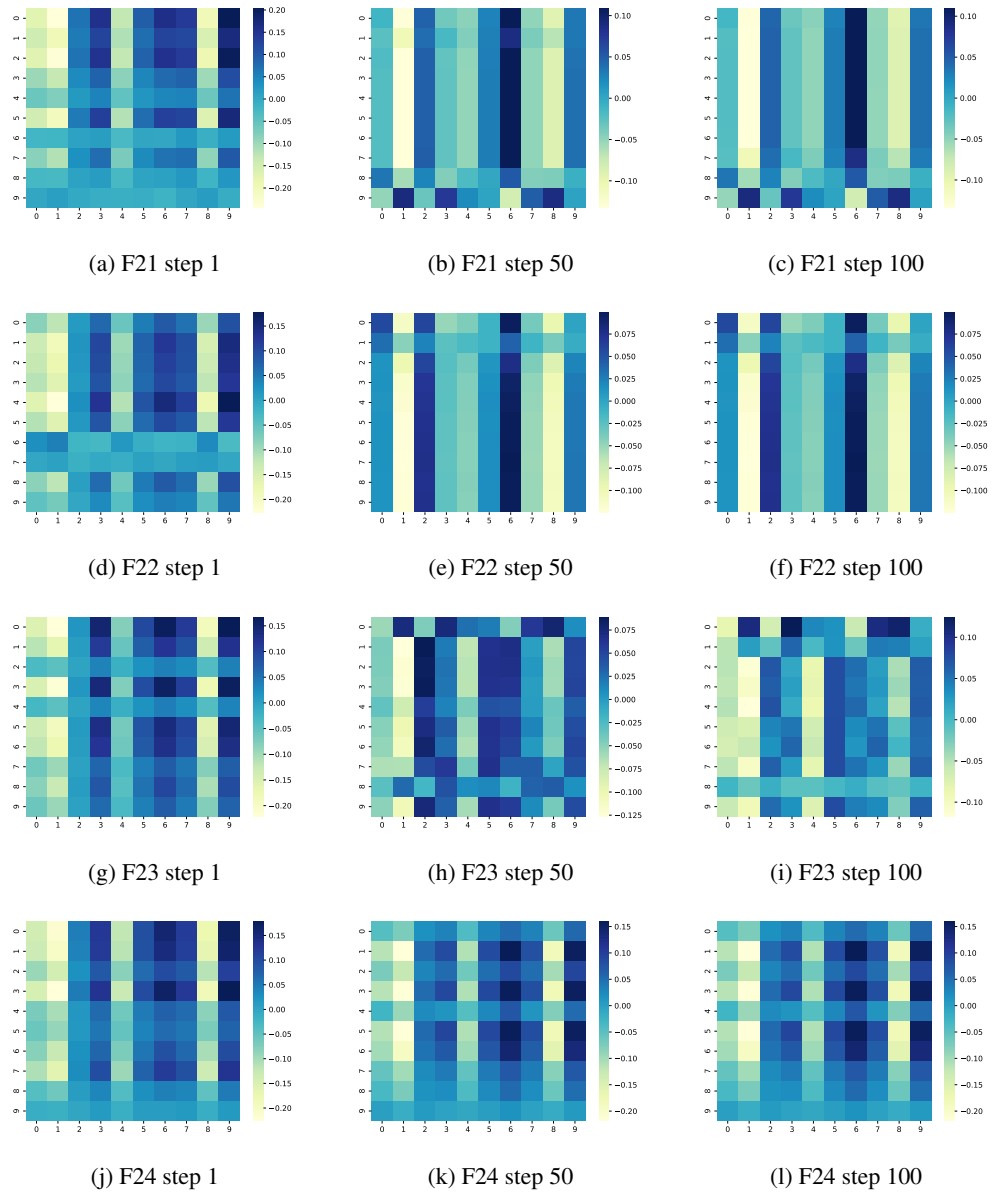

Figure 14: F21-F24 Visualization results of some information selection strategies of Efficient LMutM. 0->9 represents the population ranking, and the smaller the ranking, the higher the fitness.

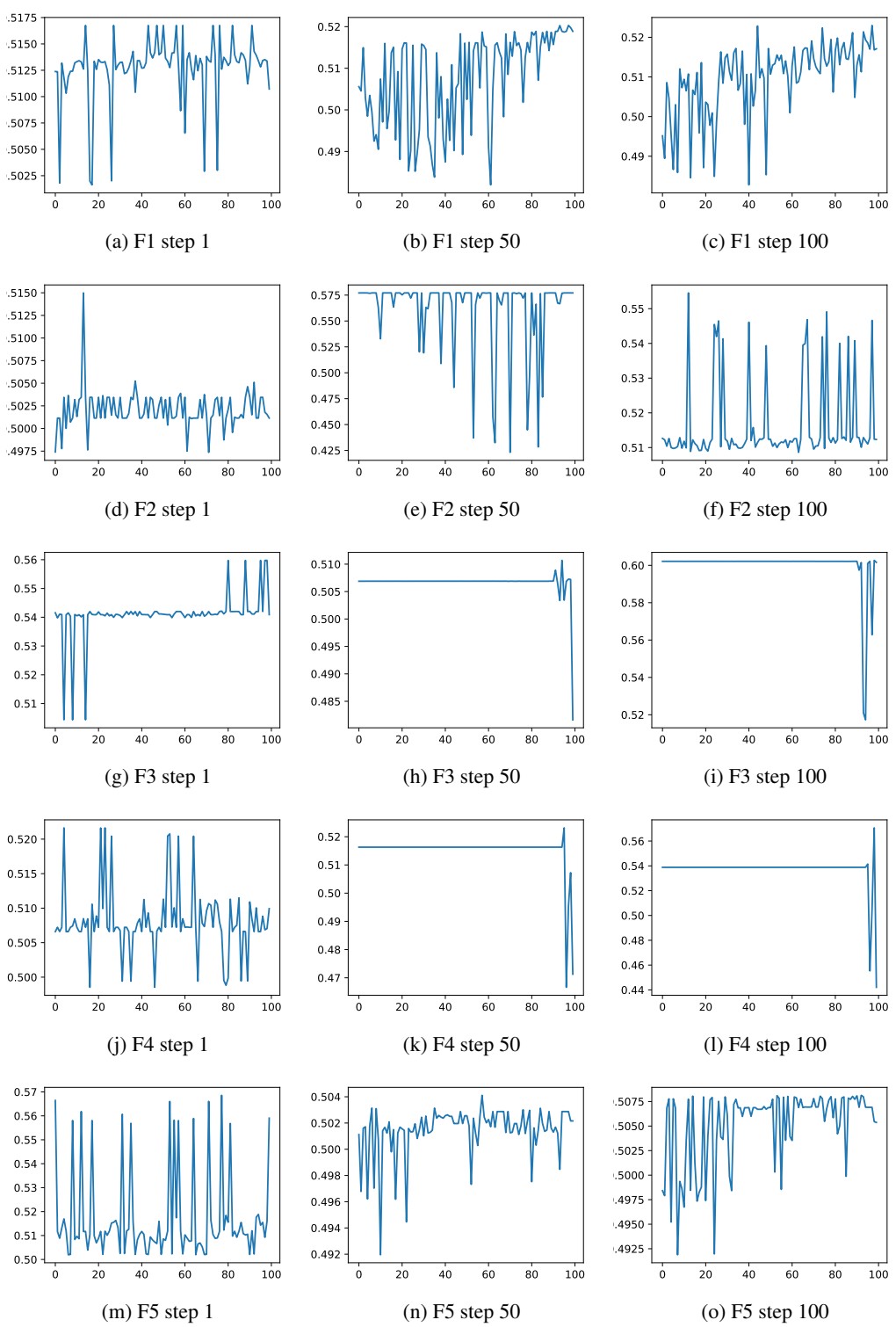

Figure 15: F1-F5 Visualization results of crossover weight of Differentiable LCrM.

# E    Visualization Results of Differentiable LCrM

The visualization results of Differentiable LCrM are shown in Figure 15-19.

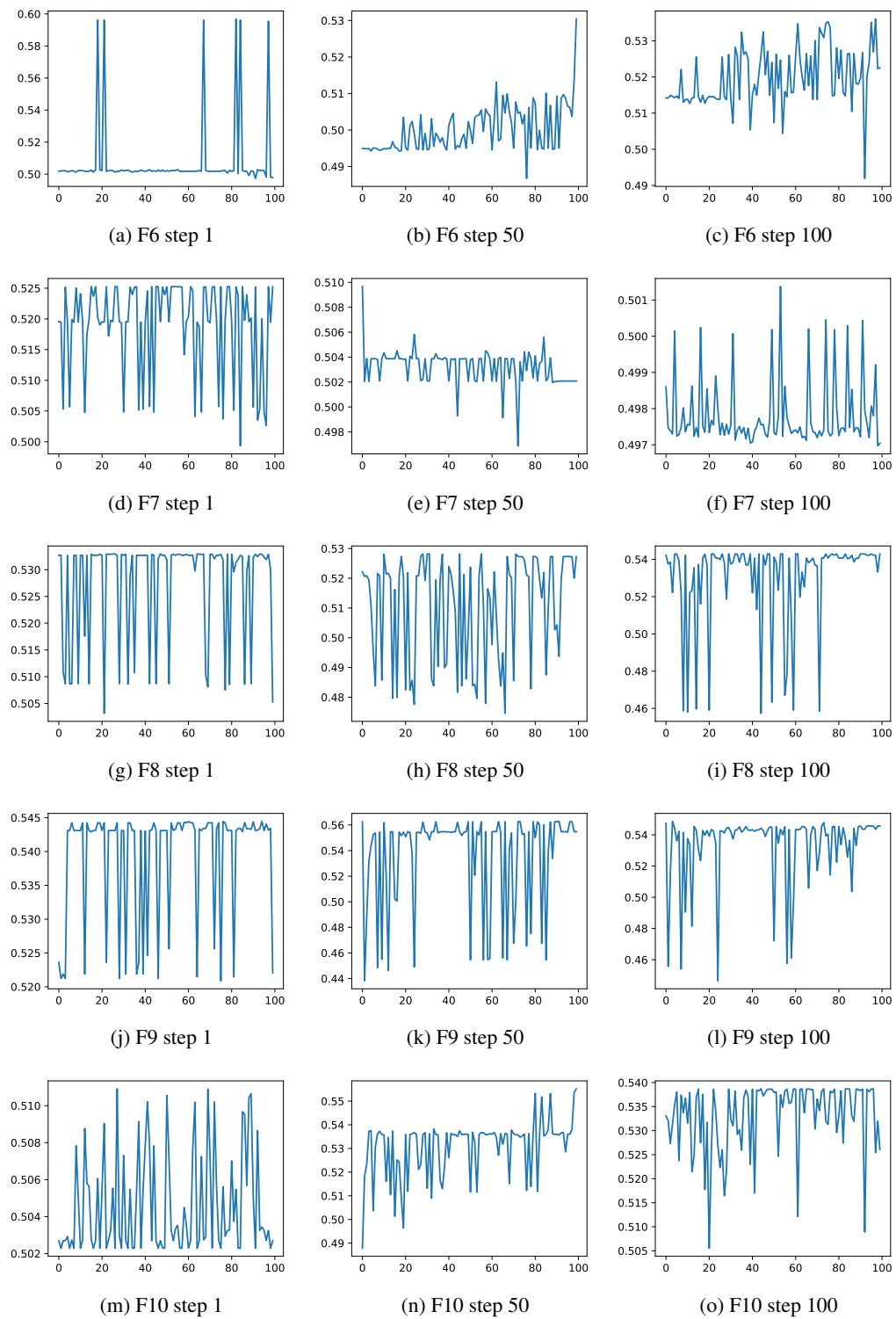

Figure 16: F6-F10 Visualization results of crossover weight of Differentiable LCrM.

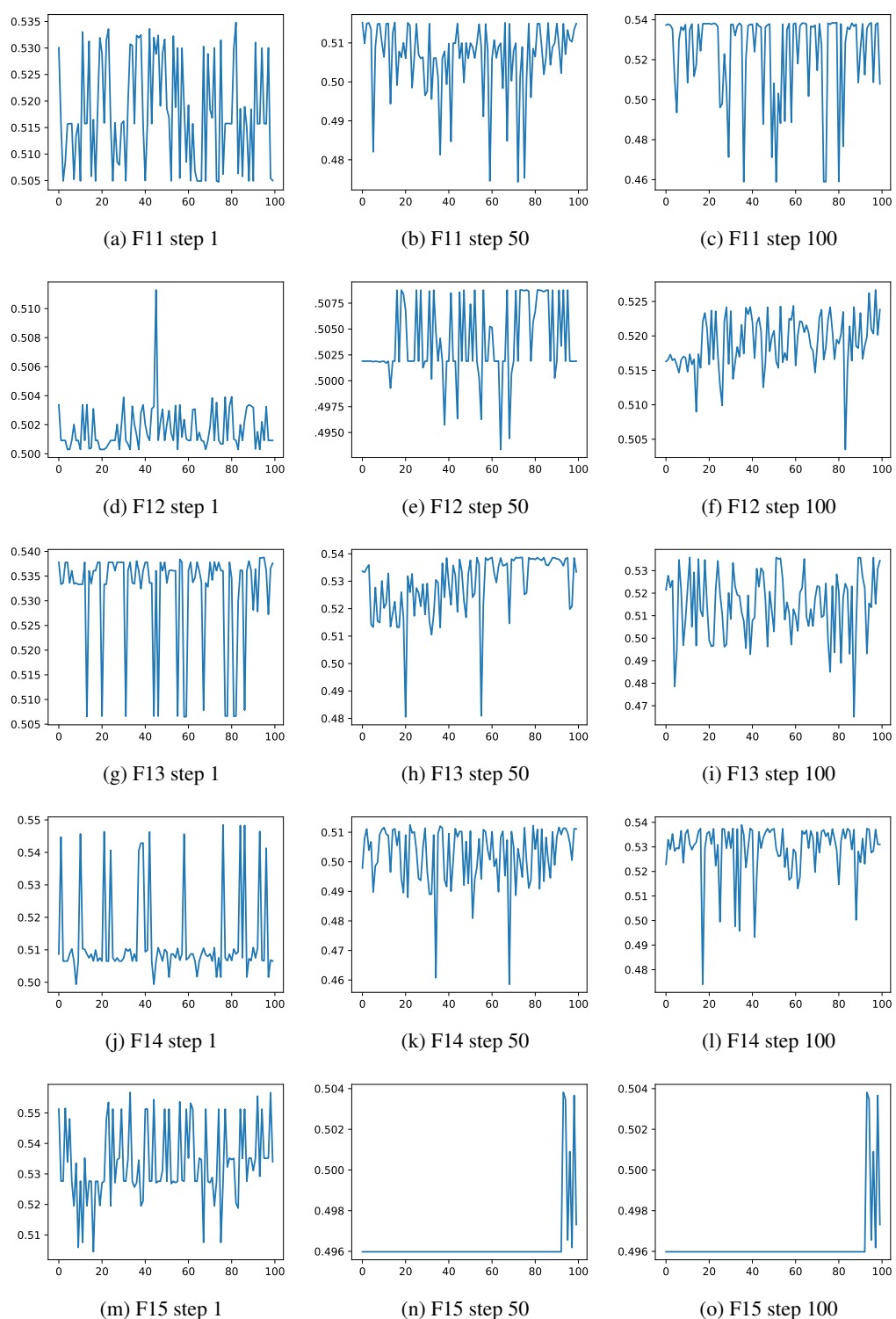

Figure 17: F11-F15 Visualization results of crossover weight of Differentiable LCrM.

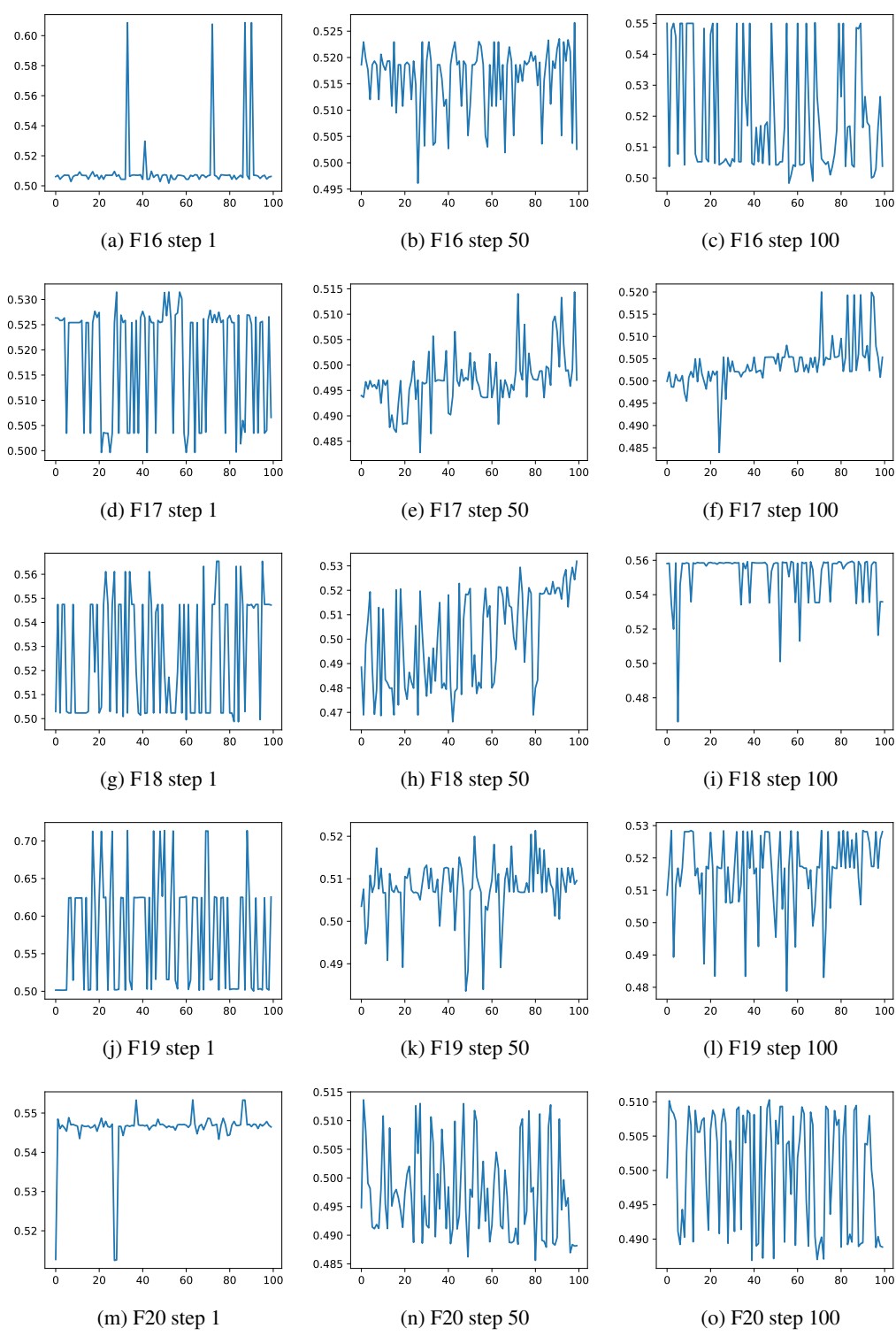

Figure 18: F16-F20 Visualization results of crossover weight of Differentiable LCrM.

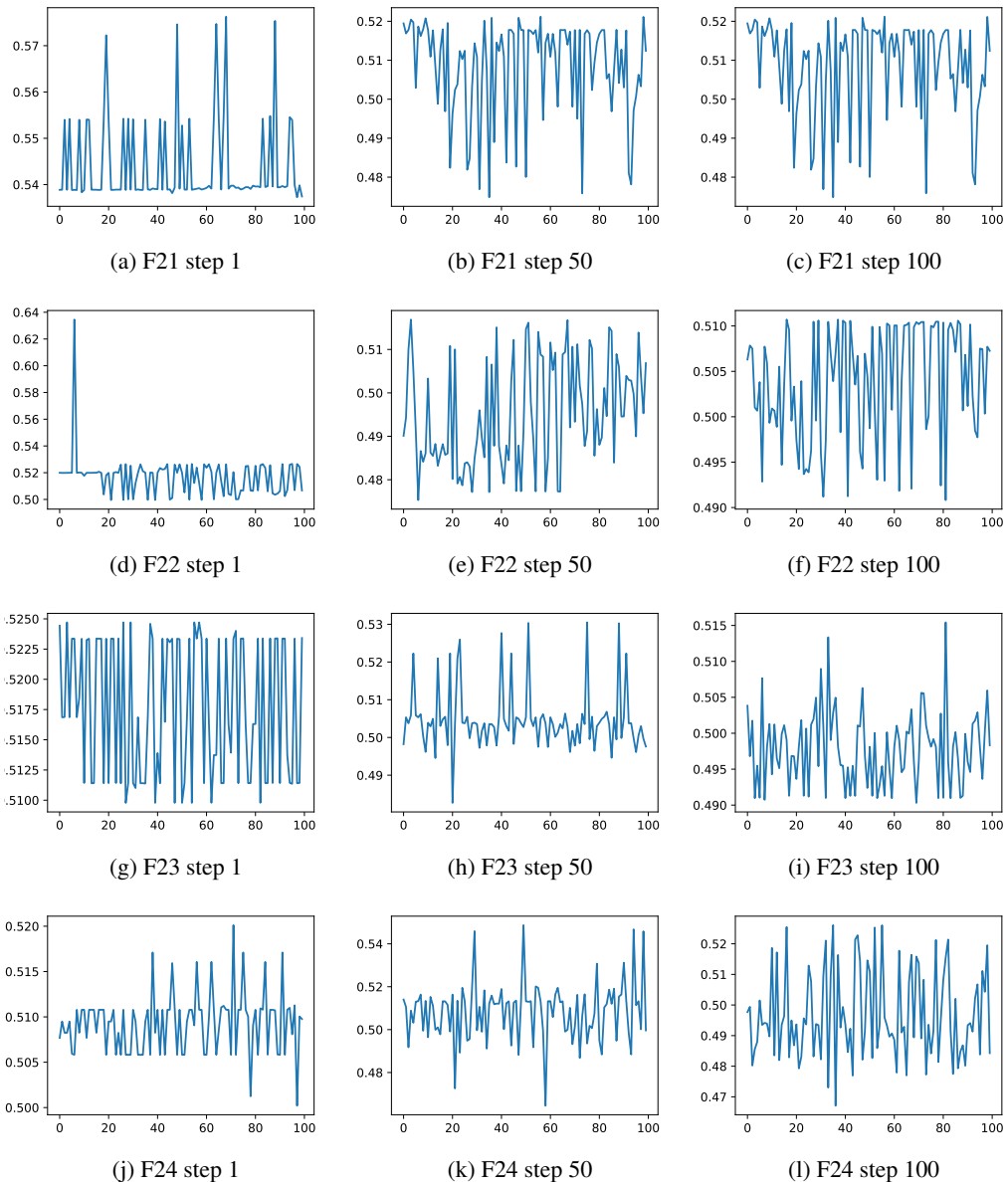

Figure 19: F21-F24 Visualization results of crossover weight of Differentiable LCrM.

