# OpenReview forum: "Enhancing Zero-Shot Black-Box Optimization via Pretrained Models with Efficient Population Modeling, Interaction, and Stable Gradient Approximation"
_NeurIPS.cc/2025/Conference — NeurIPS 2025 poster_

### Official Review · Reviewer_P2q1 · 2025-06-25

**Clarity:** 2
**Significance:** 3
**Originality:** 3
**Rating:** 4
**Confidence:** 4

**Summary:**

This paper proposes EPOM (Efficient Pretrained Optimization Models), a new architecture for zero-shot black-box optimization. It improves on previous POMs by introducing (1) richer population tokenization (LMM/LCM Tokenizers), (2) deformable attention for better adaptive interaction between individuals (Efficient LMM), and (3) Differentiable LCM for more stable training gradients. The method is extensively benchmarked on BBOB test functions and an RL locomotion task, achieving improved performance and convergence over state-of-the-art population-based optimization methods.

**Questions:**

- Can you provide more justification for not comparing EPOM to LLM-based optimization approaches? Are there settings where such methods might be competitive?

- Have you tested EPOM on real-world scientific or engineering problems (e.g., materials discovery, control design), beyond synthetic BBOB tasks?

- What is the computational cost of EPOM during training and inference, compared to POM or other methods? Is the method practical for problems with extremely large dimensions?

- Does EPOM require large-scale pretraining datasets? How would it perform in low-data regimes?

**Ethical Concerns:**

["NO or VERY MINOR ethics concerns only"]

**Final Justification:**

The authors addressed most of my concerns, so I will keep my positive score to support the acceptance of this paper.

**Limitations:**

Yes, included in the conclusion section.

**Quality:**

3

**Strengths And Weaknesses:**

Strengths

- Addresses an important and challenging problem: efficient zero-shot black-box optimization.

- Proposes a technically sound architecture with improvements over existing POM methods.

- Uses attention mechanisms and improved tokenization to enhance performance.

- Shows strong experimental results on BBOB benchmarks (d=30 and d=100).

- Provides good ablation studies and analysis of module contributions.

Weaknesses

- The innovations are mostly architectural — no fundamentally new learning paradigm.

- Experiments are limited to BBOB and a simple RL control task — lacks validation on more realistic scientific or engineering problems.

- No comparison to recent LLM-based optimizers (though the authors argue they are weaker for this problem class).

---

> ### Author Rebuttal · Authors · 2025-07-31
>
> > Weakness 1: The innovations ...
>
> Our submission first summarizes the principle of strong generalization of POM and then enhances it. The details are as follows:
>
> Their core idea is to learn the mapping between task features and optimization strategies, enabling the model to directly output high-quality optimization strategies without fine-tuning for new tasks.
>
> We summary this process into three key elements: task feature modeling, optimization strategies expression, and a mapping mechanism from task features to optimization strategies. First, the task feature set, typically consisting of normalized fitness values and centralized rankings of individuals, characterizes the structural characteristics of the optimization task. Its expressive power directly determines the POM's ability to perceive and adapt to different tasks. Second, the optimization strategies, as the output of the POM, must possess sufficient expressive power to cover search behaviors across diverse tasks. The mapping from features to strategies, learned during the pre-training phase, is key to achieving zero-shot performance.
>
> Although existing POM methods have achieved some success, significant bottlenecks remain in feature modeling, strategies expression, and training stability. To this end, this paper proposes systematic improvements around the three core elements mentioned above, building a POM system with enhanced generalization and training robustness:
>
> Enhanced Task Feature Modeling: To improve the expressiveness of task features, we propose the LMM/LCM Tokenizer. This module utilizes cross-attention to construct representations of decision variables of arbitrary dimensions and concatenates them with the original features, resulting in a richer and more structure-aware problem representation.
>
> Enhanced Optimization strategies Representation: We introduce deformable attention into the original LMM module, proposing the Efficient LMM, to enhance flexibility and information exchange during strategies generation, achieving significant performance improvements on multiple benchmark tasks.
>
> Enhanced Training Stability: To overcome the gradient estimation inaccuracies caused by Gumbel-Softmax, we design a differentiable crossover mechanism, Differentiable LCM. This replaces the original non-differentiable crossover process with a weighted summation, effectively improving the stability and convergence quality of the training process.
>
> In summary, by synergistically enhancing feature modeling, strategies representation, and mapping mechanisms, this paper promotes the development of pre-trained optimization models with stronger zero-shot optimization capabilities and wider adaptability.
>
> > W. 2: Experiments are limited ...
>
> We evaluate the performance of EPOM and POM on the planner mechanic arm problem, a engineering problem that has been widely used to evaluate the performance of BBO algorithms. The optimization goal of this problem is to search for a set of lengths $\mathbf{L} = [l_1, l_2, \cdots, l_n]$ and a set of angles $\mathbf{\alpha} = [\alpha_1, \alpha_2, \cdots, \alpha_n]$ to minimize the distance from  to the target position based on a given $r$, represents the distance from the target point to the origin of the mechanic arm . In our experiment, we let $n=30$, $r=100$, fix the length $l_i = 10$, and search for the optimal angles only.
> We used POM and EPOM to optimize the problem for 100 generations each, and recorded the best fitness (minimum distance) of the last generation. The optimal value achieved by EPOM was **2.63(1.07)**, and that of POM was 8.09(2.98).
>
> > W. 3: No comparison ...
>
>  LLM-based optimizers lack zero-shot optimization capabilities, so comparison with them is meaningless. Related work describes this capability as being related to pre-trained models.
> This may not be our weakness.
>
> > Question 1: Can you provide ...
>
> Please see our rebuttal to **Weakness 3**.
>
> > Question 2: Have you tested ...
>
> Please see our rebuttal to **Weakness 2**.
>
> > Q. 3: What is the ...
>
> We recorded the time consumption (in seconds) of EPOM, POM and CMAES when optimizing 24 BBOB functions in 30- and 100-dimensional settings respectively, as shown in the following table.
> |Method|d=30|d=100|
> |---|---|---|
> |EPOM|22.29(0.22)|23.10(0.36)|
> |POM|**4.84(0.24)**|**5.57(0.23)**|
> |CMAES|11.34(0.24)|33.99(1.99)|
> In summary, POM, thanks to its streamlined model structure, achieved the lowest time consumption in both settings. CMAES, however, lacks GPU acceleration, and while it consumes less time than EPOM in the low-dimensional setting, it consumes approximately 10.89 seconds more than EPOM in the 100-dimensional setting. For EPOM, the increased time and computational effort required for performance improvement, driven by its increased model complexity, is almost an inevitable price to pay.
>
> EPOM can generalize well across dimensions. We tested the performance of POM and EPOM on BBOB in a 300-dimensional setting and recorded their best fitness in the last generation on each function, part of the results are shown in the following table.
>
> |method |f1|f8|f15|f22|
> |---|---|---|---|---|
> |EPOM|2.88e-54(3.21e-54)|5.24e-18(7.41e-18)|3.05e-51(4.31e-51)|83.78(0.45)|
> |POM|1.40e-10(6.54e-11)|9.55e-6(5.89e-6)|1825.98(247.68)|79.65(0.54)|
>
> > Q. 4: Does EPOM ...
>
> In our submission, EPOM was trained on only five functions under the 10-dimensional setting (as shown in Table 1). Compared to the 24 BBOB functions in the 30- and 100-dimensional settings used for testing, the training set can be considered a low-data regime. Despite this, EPOM demonstrates strong performance in the experiments, indicating that it does not require a large-scale pretraining datasets.

---

> > ### Comment · Reviewer_P2q1 · 2025-08-04
> > **Reply to the rebuttal**
> >
> > Thank you for your detailed rebuttal and for taking the time to address my concerns, as well as for running additional experiments. I appreciate the clarifications and your effort to strengthen the submission. I will keep my score to support the paper's acceptance. Best wishes to the authors.

---

> > > ### Author Response · Authors · 2025-08-05
> > >
> > > **Dear Reviewer,**
> > >
> > > Thank you for your support and valuable feedback throughout the review process. We sincerely appreciate your recognition of our revisions—this means a great deal to us. We are grateful for the time and expertise you dedicated to improving our paper. Your insights have undoubtedly strengthened our work.
> > >
> > > Rest assured, we will carefully consider all points discussed during this review cycle as we work to further refine and improve the paper.
> > >
> > > **Sincerely,**
> > >
> > > **Authors**

---

### Official Review · Reviewer_X9vv · 2025-07-02

**Clarity:** 1
**Significance:** 3
**Originality:** 3
**Rating:** 5
**Confidence:** 4

**Summary:**

The work presented in this paper, titled "Enhancing Zero-Shot Black-Box Optimization...", introduces a framework referred to as EPOM. This framework is designed to improve upon a prior method, Pre-trained Optimization Models (POMs), for the task of zero-shot black-box optimization. The authors identify three primary weaknesses in the existing POM framework—namely, insufficient population modeling, inefficient information exchange, and unstable gradient approximation—and propose a corresponding set of architectural solutions. These solutions involve: (1) a new tokenizer module to create richer embeddings of the solution population; (2) a mutation operator, which they call an "Efficient LMM," based on deformable attention to facilitate adaptive interactions; and (3) a crossover operator, which they call a "Differentiable LCM," that utilizes a weighted-sum mechanism to improve the stability of the meta-training process. The authors provide an empirical evaluation on the BBOB benchmark suite, where their proposed EPOM model demonstrates superior performance compared to the baseline POM and other established optimization algorithms.

**Questions:**

My current rating is 'Borderline Reject' due to the severe issues with clarity and conceptual framing. For the paper to reach an acceptance status, the authors must undertake a major revision to address the following points. A successful revision that thoroughly resolves these issues would be necessary to reconsider my evaluation.

Rewrite for Clarity and Accessibility: The paper must be substantially rewritten. All bespoke acronyms, especially the misleading 'LMM' and 'LCM', must be clearly defined upon their first use in the Abstract and Introduction, or preferably, replaced with less ambiguous terms. The convoluted sentence structures should be simplified to make the core ideas accessible to a general machine learning audience.

Explicitly Position the Work within Meta-Learning: The paper must situate itself correctly within the meta-learning literature. The introduction and related work sections need to explicitly discuss how "Pre-trained Optimization Models" relate to and build upon decades of research in meta-learning. The authors should either justify their novel terminology or adopt the standard language of the field.

Clarify Architectural Contributions: Given the complexity of the system, a clearer ablation is needed. What is the single most impactful component? Is the performance gain primarily from the deformable attention mechanism, the tokenizer, or the differentiable crossover? A more fine-grained analysis would help distill the core engineering takeaway of the paper.

**Ethical Concerns:**

["NO or VERY MINOR ethics concerns only"]

**Final Justification:**

See my response to rebuttal for full details.

TLDR; Authors addressed most of my concerns satisfactorily and I am now happy to accept the paper.

**Limitations:**

Yes.

**Paper Formatting Concerns:**

None.

**Quality:**

2

**Strengths And Weaknesses:**

Strengths:
The primary strength of this work lies in its empirical results. The experimental evaluation is extensive and appears to be robust, showcasing that the proposed EPOM system consistently outperforms a range of strong baselines, including the original POM architecture and classical methods like CMA-ES, on a challenging benchmark suite. The engineering effort is clearly substantial, and the final model is demonstrably effective at its stated goal.

Weaknesses:
Despite the strong empirical results, the paper is beset by significant weaknesses in its presentation, clarity, and conceptual positioning, which collectively form a significant barrier to its understanding and potential impact.

Clarity and Terminology: The writing, particularly in the Abstract and Introduction, is convoluted and appears almost intentionally gatekeeping. It is laden with bespoke jargon and acronyms that are not immediately defined, making the paper exceptionally difficult to penetrate for any reader not already deeply familiar with the specific baseline paper [7]. This lack of clarity is a major flaw.

Misleading Acronyms: The use of the acronyms 'LMM' and 'LCM' for "Learned Mutation Module" and "Learned Crossover Module" is exceptionally poor practice and must be addressed. In the broader machine learning community, LMM almost universally stands for Large Multimodal Model, while LCM has been recently associated with Large Context Models or Latent Consistency Models. Defining these overloaded acronyms deep within the paper (Section 3.1.1) is insufficient. This choice is not merely confusing; it is actively misleading and suggests a lack of awareness of the wider field.

Conceptual Positioning: The work is presented as "zero-shot optimization" via "Pre-trained Optimization Models," creating an artificial distinction from the well-established field of meta-learning. A reader well-versed in meta-learning is left to wonder why this new terminology is necessary when the described process—training a model on a distribution of tasks to generalize to new tasks—is the very definition of meta-learning. This failure to connect to, or even acknowledge, the foundational concepts of the field it operates in is a critical conceptual weakness.

In summary, while the engineering is sound, the poor writing, confusing terminology, and flawed conceptual framing make the work hard to digest and will severely limit its potential impact and its ability to be integrated into the broader discourse on meta-learning.

---

> ### Author Rebuttal · Authors · 2025-07-31
>
> > Weakness 1 Clarity and Terminology ...
>
> We rewrote the Abstract and Introduction to highlight the motivation, novelty, and coherence of the paper.
>
> Abstract
> Zero-shot optimization aims to achieve both generalization and performance gains on solving previously unseen black-box optimization problems over SOTA methods without task-specific tuning. Pre-trained optimization models (POMs) address this challenge by learning a general mapping from task features to optimization strategies, enabling direct deployment on new tasks.
>
> In this paper, we identify three essential components that determine the effectiveness of POMs: (1) task feature modeling, which captures structural properties of optimization problems; (2) optimization strategy representation, which defines how new candidate solutions are generated; and (3) the feature-to-strategy mapping mechanism learned during pre-training. However, existing POMs often suffer from weak feature representations, rigid strategy modeling, and unstable training.
>
> To address these limitations, we propose EPOM, an enhanced framework for pre-trained optimization. EPOM enriches task representations using a cross-attentive tokenizer, improves strategy diversity through deformable attention, and stabilizes training by replacing non-differentiable operations with a differentiable crossover mechanism. Together, these enhancements yield better generalization, faster convergence, and more reliable performance in zero-shot black-box optimization.
>
>
> Introduction
>
> ... Traditionally, solving BBO tasks requires expert-designed or heavily tuned optimization algorithms for each new problem, making the process inefficient and non-scalable. Thus, the paradigm of solving new optimization problems without any task-specific tuning is extremely valuable.
>
> Meta-learning, or learning to learn [1], particularly in the context of meta-black-box optimization [2] or learning to optimize [3], covers a broad range of scenarios. These approaches often leverage information from the target task to enhance the optimizer’s performance on that specific task. However, they typically exhibit poor generalization to new, unseen tasks. Even though some methods—such as LES \cite{lange2023discovering1} and LGA \cite{lange2023discovering}—can generalize to novel settings, their performance still falls significantly short compared to state-of-the-art optimizers specifically designed for those settings.
>
> To better highlight our contribution, we introduce the concept of zero-shot optimization, which aims to simultaneously achieve strong generalization and high performance on new optimization problems—without requiring task-specific training. A particularly promising direction in this area is the development of pre-trained optimization models (POMs). These models are designed to learn general-purpose optimization behaviors across a wide variety of training tasks and effectively transfer that knowledge to previously unseen problems.
>
>
> We view a POM as a function that maps task-specific features to optimization strategies. This process involves three key components:
> (1) **Task feature modeling**, which captures characteristics of the problem (e.g., normalized fitness values and centralized rankings);
> (2) **Optimization strategy representation**, which determines how candidate solutions are generated and refined;
> (3) **Mapping mechanism**, which learns the transformation from features to strategies during pre-training.
>
> Although recent POMs have made progress, they remain limited in three areas:
> - The expressiveness of task features is insufficient to generalize across diverse problem structures;
> - The strategy generation process lacks flexibility and diversity;
> - The training process suffers from gradient instability due to non-differentiable operations.
>
> To overcome these limitations, we propose **EPOM (Enhanced Pretrained Optimization Model)**, which systematically enhances all three components:
> - **Feature modeling** is improved by a cross-attentive tokenizer that captures decision-variable-level information and encodes it into fixed-length task representations;
> - **Strategy representation** is enhanced through deformable attention, enabling dynamic and diversity-aware interactions among population members;
> - **Training stability** is ensured by replacing sampling-based crossover operations with a differentiable, weighted-sum mechanism, leading to smoother gradient flow.
>
> By jointly improving feature modeling, strategy generation, and training robustness, EPOM achieves superior generalization and performance in zero-shot black-box optimization. This work contributes to the growing body of meta-learning literature on learning-to-optimize methods, and offers a scalable pathway toward universal optimization agents.
>
> > W. 2: Misleading Acronyms ...
>
> Thank you for your suggestion. We have revised the two terms to: Learned Mutation Module (LMuM) and Learned Crossover Module (LCrM) to distinguish them from the existing concepts.
>
> > W. 3: Conceptual Positioning ...
>
> We thank the reviewer for raising this important point. While our work is indeed related to meta-learning, we use the term zero-shot optimization to emphasize a more specific setting: directly generating optimization strategies for unseen tasks without task-specific adaptation, based solely on task features.
>
> Meta-learning, particularly meta-black-box optimization [1], encompasses broader scenarios such as algorithm selection, configuration, and solution manipulation. These settings do not align exactly with our goal of optimization strategy generation with strong generalization and superior performance. Similarly, terms like learning to optimize often imply inner-loop adaptation, which our method avoids.
>
> Thus, introducing the term zero-shot optimization helps clarify our contribution: achieving both generalization and performance gains over SOTA methods without per-task training. We explicitly define this term in the introduction (Lines 25–27) and compare with meta-learning baselines in both related work and experiments, where our method demonstrates superior effectiveness.
>
> Therefore, the use of this terminology is intentional and necessary, rather than a conceptual flaw.
>
> [1] Hospedales, T., Antoniou, A., Micaelli, P., & Storkey, A. (2021). Meta-learning in neural networks: A survey. IEEE transactions on pattern analysis and machine intelligence, 44(9), 5149-5169.
>
>
> [2] Z. Ma, H. Guo, Y. -J. Gong, J. Zhang and K. C. Tan, "Toward Automated Algorithm Design: A Survey and Practical Guide to Meta-Black-Box-Optimization," in IEEE Transactions on Evolutionary Computation, doi: 10.1109/TEVC.2025.3568053.
>
> [3] Chen, T., Chen, X., Chen, W., Heaton, H., Liu, J., Wang, Z., & Yin, W. (2022). Learning to optimize: A primer and a benchmark. Journal of Machine Learning Research, 23(189), 1-59.
>
> > Question 1: Rewrite for ...
>
> Based on your suggestions, we will simplify complex sentences.
>
> > Q. 2: Explicitly Position ...
>
>  In the introduction, we discussed the differences and connections between pre-trained optimization base models and meta-learning. For details, see the second and third paragraphs of the Introduction section.
>
> > Q. 3: Clarify Architectural Contributions ...
>
> We thank the reviewer for their suggestions. We will enrich the ablation study in the revision and conduct in-depth analysis of each module.

---

> > ### Author Response · Authors · 2025-08-07
> >
> > We sincerely thank you for your initial review and the constructive comments. We have carefully addressed all the concerns in our rebuttal. As the discussion phase will conclude in 36 hours, we would greatly appreciate it if you could kindly take a moment to review our response and share any additional thoughts. Your input is invaluable to us, and we are grateful for your time and consideration.
> >
> > Sincerely,
> >
> > Authors

---

### Official Review · Reviewer_NPVS · 2025-07-03

**Clarity:** 2
**Significance:** 3
**Originality:** 3
**Rating:** 5
**Confidence:** 3

**Summary:**

This paper proposes Efficient Pretrained Optimization Models (EPOM), a method for solving black-box optimization problems without task-specific training. EPOM builds off of prior Pretrained Optimization Models (POM) in a few ways: The addition of an LMM/LCM tokenizer which embeds the population into fixed-length tokens to allow the model to capture more landscape cues, a deformable attention module for the LMM, and replacing the simple gumbel softmax LCM with a learned module that has stable gradients. Evaluation on a suite of black box problems show better performance and convergence compared to prior methods.

**Questions:**

1. Could you clarify the results in table 2?

The paper makes multiple claims about how EPOM is especially good with higher dimensions, d=100, but table 2 seems to suggest that it's worse in this case. In fact, it does worse than POM more often than not for d=100. Furthermore, there seems to be a miss match between the plots in figures 6-9 of the appendix and table 2 mentioned previously. I lean reject with these issues outstanding, but I'm willing to increase my recommendation based on the authors response.

2. How much deviation from a crossover weight of 0.5 is significant?

Figure 5(c)(d) shows very little deviation in the crossover weight (0.49-0.52), but the text claims that this is significant. I would imagine replacing this with a static 0.5 would lead to very similar performance. Looking at the appendix, there do seem to be cases where the crossover weight has much more variance, but these examples weren't selected for figure 5. Is there a pattern we can draw for when the crossover weight is close to 0.5 vs when it is not?

3. How does the runtime of EPOM compare to POM?

The appendix discusses the compute requirements, but doesn't say how this compares to POM. Does EPOM require much more or less compute than POM?

**Ethical Concerns:**

["NO or VERY MINOR ethics concerns only"]

**Final Justification:**

My main concern with this paper was with unclear results in both the main table and appendix. The authors provided clarifications to my main concerns, which better explain the results and alleviate my worry. They also provide additional analysis on runtime and crossover weight which further justifies the method. For these reasons I now recommend acceptance.

**Limitations:**

yes

**Paper Formatting Concerns:**

There might be an equation missing in 3.1.1 part 2). There seems to be a description of terms but they aren't used in any equation that I see.

I think figure 4 would be more clear with a simple table.\

**Quality:**

3

**Strengths And Weaknesses:**

The overall quality is good, the plots are well made with nice error bars and figure 1 is clear. Figure 4 is a bit confusing with small text and overlapping lines, I think a simple table would work better, an also a better explanation of which function is being optimized.

The clarity seems fair. The methods section is very clear with the description of each component and loss/bbox functions. There are a few things that seem unclear. First, there seems to be some discrepancies with figures 6-9 in the appendix and table 2. For example, f4 with d=100 shows better performance for POM in table 2, but has EPOM doing better in figure 8 (d). Additionally, section 4.2 claims EPOM is especially good with d=100, but table 2 shows that it's more often worse than POM for d=100 (POM is better 13/24 tasks). Finally, the crossover weight doesn't deviate much from 0.5 but the text doesn't justify if this is significant.

Overall the significance also seems good. EPOM seems to converge much better in many problems which is promising, however it adds much more complexity and possibly compute. I have some concerns with the results mentioned previously that make me unsure if conclusions drawn are accurate.

The originality is also good. There are many novel adjustments made to POM and they are justified with ablations. The performance improvements would certainly be of interest to the larger community.

---

> ### Author Rebuttal · Authors · 2025-07-31
>
> > Weakness 1: Figure 4 is a bit confusing ...
>
> We hope the following explanation will help the reviewer better understand Figure 4. First, Figure 4 is a critical difference diagram, which visualizes the performance differences among the given algorithms on specified datasets based on their average rankings. The results of our ablation study are summarized in this figure.
> Specifically, we remove each proposed module one by one and evaluate the corresponding variant of EPOM on 24 BBOB functions in 30 dimensions. For each function, we compute the ranking of all variants and then calculate the average rank, which is shown as the number above each polyline in Figure 4.
> We suspect that what the reviewer referred to as “overlapping lines” mainly concerns the lines representing the “No Differentiable LCM” and “No LMM Tokenizer” settings. This overlap is due to their similar average rankings. To address this issue, we will increase the figure’s size and resolution in the next revision, which should alleviate the visual ambiguity.
>
> Additionally, following the reviewer’s suggestion, we have made the following table based on the average rankings in Figure 4, which we hope will further clarify the results and assist in interpreting the figure.
> | Settings | Average Rank |
> |----------|--------------|
> | EPOM | 1.6250 |
> | No LCM Tokenizer | 1.8750 |
> | No Differentiable LCM | 3.2500 |
> | No LMM Tokenizer | 3.2917 |
> | No Efficient LMM | 4.9583 |
> | No SM | 6.0000 |
>
> > W. 2: The clarity seems fair ...
>
> **(a)** We sincerely apologize for our oversight and the confusion it may have caused the reviewer. The discrepancies between Table 2 and Figures 6–9 are mainly caused by their different calculation methods. For each entry $m(n)$ in Table 2, $m$ and $n$ represent the mean and standard deviation of the fitness of the full population, for a given function, method, and dimension setting. Fig. 6-9 represent the optimization trajectory curves consisting of the best fitness in each generation. Once again, we sincerely apologize for not describing these two experimental setups clearly in our submission, and for any confusion this may have caused.
>
> **(b)** Following **(a)**, our conclusion are drawn based on the best fitness of the last generation of each method, because the core goal of black-box optimization algorithms is to find the optimal solution as much as possible, rather than to make all solutions close to the optimal. The best fitness of the last generation can better reflect both this goal and the best performance of a given algorithm.
>
> **(c)** We tested EPOM with constant crossover weight(0.5) and varying crossover weight on the BBOB in a 30-dimensional setting. On some functions, their performance is not much different, but on some functions, their performance is quite different, as shown in the following table (the best fitness of the last generation is recorded).
>
> ||f1|f11 |f16|
> |---|---|---|---|
> |cr=0.5|1.69e-55(1.90e-55) |5.55e-39(7.85e-39) |21.99(3.60)|
> |varying cr|8.85e-54(5.45e-54) |4.24e-50(2.81e-50)|32.55(2.30)|
>
> > W. 3: however it adds much more complexity ...
>
> We recorded the time consumption (in seconds) of EPOM, POM and CMAES when optimizing 24 BBOB functions in 30- and 100-dimensional settings respectively, as shown in the following table.
> ||30|100|
> |---|---|---|
> |EPOM|22.29(0.22)|23.10(0.36)|
> |POM|**4.84(0.24)**|**5.57(0.23)**|
> |CMAES|11.34(0.24)|33.99(1.99)|
>
> In summary, POM, thanks to its streamlined model structure, achieved the lowest time consumption in both settings. CMAES, however, lacks GPU acceleration, and while it consumes less time than EPOM in the low-dimensional setting, it consumes approximately 10.89 seconds more than EPOM in the 100-dimensional setting. For EPOM, the increased time and computational effort required for performance improvement, driven by its increased model complexity, is almost an inevitable price to pay.
>
> > Question 1: Could you clarify ...
>
> Please see our rebuttal to Weakness 2 **(a)** and **(b)**.
>
> > Q. 2: How much deviation ...
>
> Please see our rebuttal to Weakness 2 **(c)**.
>
> > Q. 3: How does the runtime ...
>
> Please see our rebuttal to Weakness 3.
>
> > Paper Formatting Concern 1: There might be ...
>
> We sincerely thank the reviewer for carefully identifying this issue. We acknowledge that some equations related to variables such as $\mathbf{cv}^t_i$ were regrettably omitted in the current version. We provide the missing equations below:
> $$
> \mathbf{cr}^t \leftarrow LCM(\mathbf{Z}^t) \\
> \mathbf{r}^t _ i = rand(d), \; \; \mathbf{cv} _ i^t = gumbel\_softmax(cat(\mathbf{r}^t _ i, tile(cr _ i^t, d))) \\
> \mathbf{u}^t _ i = \mathbf{cv} _ {i, 0}^t \cdot \mathbf{x} _ i^t + \mathbf{cv} _ {i, 1}^t \cdot \mathbf{v} _ i^t
> $$
> We will make sure to correct them in the next revision.
>
> > P. F. C. 2: I think figure 4 ...
>
> Please see our rebuttal to Weakness 1.

---

> > ### Author Response · Authors · 2025-08-07
> >
> > We sincerely thank you for your initial review and the constructive comments. We have carefully addressed all the concerns in our rebuttal. As the discussion phase will conclude in 36 hours, we would greatly appreciate it if you could kindly take a moment to review our response and share any additional thoughts. Your input is invaluable to us, and we are grateful for your time and consideration.
> >
> > Sincerely,
> >
> > Authors

---

> > ### Comment · Reviewer_NPVS · 2025-08-07
> >
> > I thank the authors for their detailed rebuttal which addresses my main concerns. I appreciate the clarifications to improve the paper’s clarity, along with the additional analysis on runtime and crossover weight. In light of these improvements, I am raising my score and recommend acceptance.

---

> > > ### Author Response · Authors · 2025-08-08
> > >
> > > Thank you for your support and valuable feedback throughout the review process. We sincerely appreciate your recognition of our revisions and your decision to raise the score—this means a great deal to us. We are grateful for the time and expertise you dedicated to improving our paper. Your insights have undoubtedly strengthened our work.
> > >
> > > Sincerely,
> > >
> > > Authors

---

### Official Review · Reviewer_gTGN · 2025-07-03

**Clarity:** 3
**Significance:** 4
**Originality:** 4
**Rating:** 5
**Confidence:** 5

**Summary:**

This paper proposes the Efficient Pre-trained Optimization Models which embeds landscape cues into population tokens for richer modeling and seamless feature fusion, and introduces the deformable attention module to enable adaptive, diversity-promoting interactions. Experimental results demonstrate superior performance over both traditional BBO methods and existing pre-trained algorithms.

**Questions:**

1. What is the meaning and value of $\bar{d}$ in Section 3.2.2? How to shape and reshape the solutions?
2. In Section 3.2.3, what is the meaning of prob(x)? How to use the top-p sampling to obtain the $n \times k$ indices in $N_t$?
3. It seems that $m_i$ is different across $x_i$, how to determine the value of $m_i$ and why is they different?
4. How to calculate $S^t$ using $N_t$?
5. Since solutions are directly tokenized, can EPOM generalize across problems with different search spaces?
6. The population size n and $d_{out}$ determine the shape of some of the network weights, they should be known values before training and testing. Why do lines 181-183 suggest uncertainty between $d_{out}$ and n?
7. What is EPOM’s time efficiency compared to traditional BBO methods and pre-trained methods?

**Ethical Concerns:**

["NO or VERY MINOR ethics concerns only"]

**Final Justification:**

The authors provided a through rebuttal that addressed my concerns. I keep my original assessment toward acceptance.

**Limitations:**

1. Though promising in the generalization across problem types and dimensions, EPOM might not be generalizable across population sizes.
2. EPOM significantly reduces the required expert knowledge in BBO optimization, however, it might introduce more computational complexity nd slow down the optimization.

**Quality:**

3

**Strengths And Weaknesses:**

Strengths:
1. The proposed method efficiently embeds the optimization population and generates the next population using neural networks without handcrafted optimization strategies. This significantly reduces reliance on human effort and expert knowledge in algorithm design.
2. The proposed loss function could balance the population diversity and convergence which is essential for effective optimization.
3. EPOM demonstrates strong generalization ability across both synthetic tasks and neuroevolution tasks, as well as problems of varying dimensions.

Weaknesses:
1. The related work could be richer. Surveys for meta-learned BBO algorithms [1][2] could enrich the paper. Recent meta-learned algorithm ConfigX [3] and LLM for optimization method LLaMEA [4] are also worth mentioning. Besides, including NeurELA [5] which shares the similar idea of using Attention to embed population would also benefit the paper.
2. The shapes of some network weights depend on population size n, hindering EPOM's generalization across different population sizes.
3. In the Tokenizer, the 2-dimensional $H^t$ is concatenated with the $d_T \times d_{1V}$-dimensional $E^t$ which has significantly higher dimension than $H^t$. It might dilute the contribution of $H^t$.


Other issues:
1. The reference for Q-Mamba links to MetaBox instead of its intended source, it might be a mistake.
2. Section 3.1.1 2) LCM appears to lack an equation that utilizes introduced variables. Additionally, the shape of $W_{2V}$ in line 180 is incompatible for multiplication with $G_t$. In Equation (3) $K_{1M}$ might need a transpose.
3. In Algorithm 1, line 9, $V^t$ should be a parameter of LCM().

Suggested references:

[1] Ma, Zeyuan, et al. "Toward automated algorithm design: A survey and practical guide to meta-black-box-optimization." IEEE Transactions on Evolutionary Computation (2025).

[2] Yang, Xu, et al. "Meta-Black-Box optimization for evolutionary algorithms: Review and perspective." Swarm and Evolutionary Computation 93 (2025): 101838.

[3] Guo, Hongshu, et al. "Configx: Modular configuration for evolutionary algorithms via multitask reinforcement learning." Proceedings of the AAAI Conference on Artificial Intelligence. Vol. 39. No. 25. 2025.

[4] van Stein, Niki, and Thomas Bäck. "Llamea: A large language model evolutionary algorithm for automatically generating metaheuristics." IEEE Transactions on Evolutionary Computation (2024).

[5] Ma, Zeyuan, et al. "Neural exploratory landscape analysis for meta-black-box-optimization." The Thirteenth International Conference on Learning Representations. 2025.

---

> ### Author Rebuttal · Authors · 2025-07-31
>
> > Weakness 1: The related work ...
>
> We appreciate the author's comment and will enrich our Related Work section according to the advice.
>
> > W. 2 The shapes of ...
>
> We believe that the "some network weights" mentioned by the reviewer refer to the attention network used to identify the individuals to be focused on, as described in Section 3.2.3. In fact, for the following reasons, we set the output dimension $d_{out}$ of such networks to a fixed value independent of the population size (as stated in the submission: "... and $d_{out}$ is the fixed output dimension.", and we set $d_{out}$ to 100 in this submission):
> (1) To avoid additional computational overhead. When the population size $n > d_{out}$, we identify the individuals to be focused on only from the first $d_{out}$ candidates in the population, thereby avoiding any increase in the computational burden of deformable attention. When $n < d_{out}$,  we clip the network output and retain only the first n dimensions. This reason should also help address the reviewer's concerns in Question 6 and Limitation 1.
> (2) To preserve generalization ability. As the reviewer pointed out, if  $d_{out}$ were determined by the population size $n$, EPOM's generalization ability across different population sizes would be compromised. By setting it to a fixed value, we mitigate this issue to some extent.
>
> > W. 3: In the Tokenizer, ...
>
> We also took this issue into consideration when designing the LMM/LCM Tokenizer. Therefore, we suggest that both $d_T$ and $d_{1V}$ should not be set too large. In this submission, we set $d_T = d_{1V} = 4$ , resulting in a 16-dimensional decision variable feature. When concatenated with the original 2-dimensional feature, its contribution would not be overly diluted.
>
> > Other Issues 1: The reference for ...
>
> We thank the reviewer for pointing out this citation issue. We will correct it in the next revision.
>
> > O. I. 2: Section 3.1.1 2) ...
>
> We sincerely thank the reviewer for carefully identifying this issue. We acknowledge that some equations related to variables such as $\mathbf{cv}^t _ i$ were regrettably omitted in the current version. We provide the missing equations below:
> $$
> \mathbf{cr}^t \leftarrow LCM(\mathbf{Z}^t) \\
> \mathbf{r}^t _ i = rand(d), \; \; \mathbf{cv} _ i^t = gumbel\_softmax(cat(\mathbf{r}^t _ i, tile(cr _ i^t, d))) \\
> \mathbf{u}^t _ i = \mathbf{cv} _ {i, 0}^t \cdot \mathbf{x} _ i^t + \mathbf{cv} _ {i, 1}^t \cdot \mathbf{v} _ i^t
> $$
> We thank the reviewer for pointing out these two writing errors. First, the dimension of $\\mathbf{W}_{2V}$ should be $(\hat{d}_t, d_{out})$ , not $(n, d_{out})$. Then, we indeed missed the transpose operation on $\mathbf{K}_{1M}$
> in the attention computation in Equation (3).
>
> We sincerely thank the reviewer once again for pointing out issues (a) and (b). We will make sure to correct them in the next revision.
>
> > O. I. 3: In Algorithm 1 ...
>
> To clarify,  $\mathbf{V}^t$ is not an input parameter of Differentiable LCM. Instead, it is processed by the LCM Tokenizer, which converts  $\mathbf{V}^t$ into  $\mathbf{F}^t$. The Differentiable LCM then operates on  $\mathbf{F}^t$ and  $\mathbf{E}^t$ to generate the offspring population  $\mathbf{U}^t$. Thus, while  $\mathbf{U}^t$ is indeed an input to the overall LCM pipeline, it is not directly used by Differentiable LCM.
>
> > Questions 1: what is ...
> In this submission, we define $\bar{d} = \lceil \frac{d}{2} \rceil$. The operations of shaping and reshaping a solution are straightforward and are detailed as follows:
> 1. Padding (if necessary):
> If the dimensionality of the solution vector $\mathbf{x}$ is odd, we append a scalar 0 to its end to obtain a new vector $\hat{\mathbf{x}}$ of length $d+1$; otherwise, we set $\hat{\mathbf{x}} = \mathbf{x}$
> 2. Shaping into a matrix:
> We then shape $\hat{\mathbf{x}}$, which is of length $2 \cdot \bar{d}$, into a $2 \times \bar{d}$ matrix $\hat{\mathbf{K}}_T$.
>
> for example, consider a 5-dimensional solution vector:
> $\mathbf{x} = [0.3, 1, 4, 9, 7]$
> Since $d=5$ is odd, we append 0 to obtain:
> $\hat{\mathbf{x}} =[0.3, 1, 4, 9, 7, 0]$
> Then we apply the $shape$ operation:
> $$
> \hat{\mathbf{K}}_T = shape(\hat{\mathbf{x}}) =
> \begin{eqnarray}
> \begin{bmatrix}
> 0.3 & 1 & 4 \\
> 9 & 7 & 0 \\
> \end{bmatrix}
> \end{eqnarray}
> $$
> The $reshape$ operation is simply the inverse, i.e., flattening $\hat{\mathbf{K}}_T$ row-wise to recover $\hat{\mathbf{x}}$.
> Both $shape$ and $reshape$ operations can be efficiently implemented in Python using tensor manipulation methods, such as $view()$ in PyTorch.
>
> > Q. 2: In Section 3.2.3 ...
>
> We clarify the meaning of $prob(\mathbf{x}^t_i)$ and the process of top-$p$ sampling as follows:
> For a given individual $\mathbf{x}^t_j$, we define $prob(\mathbf{x}^t_i)=\hat{\mathbf{N}}_{t, (j, i)}$ as the importance score (i.e., probability) of $\mathbf{x}^t_i$ being selected as a focused-on individual from the perspective of $\mathbf{x}^t_j$. The full matrix $\hat{\mathbf{N}}_t$ stores all such pairwise importance scores between individuals in the population.  To obtain the final index matrix $\mathbf{N}_t$, we perform the following 3 steps for each individual:
>
> 1. Sorting:
> For each row of $ \hat{\mathbf{N}}_t$, which corresponds to one individual, we sort all other individuals in descending order of importance scores.
> 2. Top-$p$ sampling:
> For the $j$-th individual, we progressively select top-ranked individuals until the cumulative score exceeds a predefined threshold $p$. Since the importance distributions differ across individuals, the number of selected individuals $m_j$ will also vary. $m_j$ is calculated by the equation below.
> $$
> \mathbf{\alpha} _ j = sort(\hat{\mathbf{N}} _ {t, (j, :)}) \quad \text{(in descending order)} \\
> m_j = \min \{ m \;|\; \sum_{i=1}^{m} \mathbf{\alpha} _ j[i] > p \}
> $$
>
> 3. obtaining $\mathbf{N} _ t$:
> Let $k = max\{m _ j\}^n _ {j=1}$. We collect the top-$k$ indexes in $\mathbf{\alpha} _ j$ for each individual, and stack them to form the final index matrix, $\mathbf{N} _ t \in \mathbb{R}^{n \times k}$.
>
> We hope this explanation also addresses the reviewer’s concern raised in Question 3.
>
> > Q. 3: It seems that ...
> Please see our rebuttal to Question 2.
>
> > Q. 4:
>
> We should clarify that $\mathbf{N} _ t$ is used to calculate $\mathbf{V}^t$, instead of $\mathbf{S}^t_i$. We would like to explain this process in individual-wise.
> Given the  index vector $\mathbf{N} _ {t, (j, :)}$, corresponding to the top-$k$ focused individuals of $\mathbf{x}^t _ j$, as well as the full population $\mathbf{X}^t$,  and their features $\mathbf{E}^t$. Firstly, we collect the focused-on features $\mathbf{E}^t _ j \in \mathbb{R}^{k \times \hat{d}_t}$ according to indexes in $\mathbf{N} _ {t, (j, :)}$. Then, in a similar way with LMM in POM, we obtain the query vector $\mathbf{q} _ j$, key matrix $\mathbf{K} _ j$ by $\mathbf{e} _ j$ and $\mathbf{E}^t _ j$, respectively, and calculate $\mathbf{S}^t _ i \in \mathbb{R}^{1 \times k}$. Finally, we collect $\mathbf{X}^t _ j \in \mathbb{R}^{k \times d}$ in the same way as $\mathbf{E}^t _ j$, and calculate $\mathbf{v}^t _ j =\mathbf{S}^t _ i \times  \mathbf{X}^t _ j$.
>
> > Q. 5
>
> As showed in lines 261–265 of our submission, EPOM does demonstrate a certain degree of generalization capability across numerical search spaces. However, its ability to generalize across problems involving different data types remains an open question. We believe that the LMM/LCM Tokenizer has the potential to support such generalization, and this is indeed a promising direction for future research. We thank the reviewer for highlighting this important point and would be glad to further discuss and explore it in future work.
>
> > Q. 6
>
> Please see our rebuttal to Weakness 2.
>
> > Q. 7
>
> We recorded the time consumption (in seconds) of EPOM, POM and CMAES when optimizing 24 BBOB functions in 30- and 100-dimensional settings respectively, as shown in the following table.
> ||30|100|
> |---|---|---|
> |EPOM|22.29(0.22)|23.10(0.36)|
> |POM|**4.84(0.24)**|**5.57(0.23)**|
> |CMAES|11.34(0.24)|33.99(1.99)|
> In summary, POM, thanks to its streamlined model structure, achieved the lowest time consumption in both settings. CMAES, however, lacks GPU acceleration, and while it consumes less time than EPOM in the low-dimensional setting, it consumes approximately 10.89 seconds more than EPOM in the 100-dimensional setting. For EPOM, the increased time and computational effort required for performance improvement, driven by its increased model complexity, is almost an inevitable price to pay.
>
> > Limitation 1
>
> Please see our rebuttal to Weakness 2.
>
> > Lim. 2
>
> Please see our reply to **Question 7**.

---

### Official Review · Reviewer_xpkG · 2025-07-05

**Clarity:** 3
**Significance:** 3
**Originality:** 2
**Rating:** 4
**Confidence:** 4

**Summary:**

This paper addresses the problem of zero-shot optimization by identifying three limitations in existing Pretrained Optimization Models (POMs): (1) limited population modeling, (2) static information exchange, and (3) noisy gradient estimates. To tackle these challenges, the authors propose a new framework called Efficient Pretrained Optimization Models, composed of three components: LMM/LCM Tokenizer, Efficient LMM, and Differentiable LCM. Experimental evaluations are conducted to assess the effectiveness of the proposed approach.

**Questions:**

Please refer to Weakness.

**Ethical Concerns:**

["NO or VERY MINOR ethics concerns only"]

**Final Justification:**

The rebuttal solves my concerns. I raised the related scores.

**Limitations:**

Please refer to Weakness.

**Quality:**

3

**Strengths And Weaknesses:**

Strengths:

1. The paper is generally well-written and easy to read at the sentence level.
2. The identified challenges (e.g., noise, representation, and information flow) are relevant and important for improving zero-shot optimization performance.

Weakness:

1. Unclear Problem Definition and Background. The Introduction section lacks a clear and precise formulation of the research question. Readers unfamiliar with pretrained optimization may struggle to understand what exactly is being addressed. A more structured introduction that defines the zero-shot optimization task, its applications, and the role of POMs would significantly improve clarity.
2. The motivation is unclear and unconvincing. The authors outlined three limitations of existing works in Pre-trained Optimization Models. However, the motivation for addressing the three limitations is not well-grounded. The current descriptions use abstract and general terms (e.g., "distribution characteristics," "convergence," "generalizability," "approximate gradients") without clearly explaining their practical meaning or impact. For instance, the authors mentioned that "POMs often fail to adequately capture the distribution characteristics of the population, which undermines the convergence and generalizability of the derived optimization strategies." In their statements, they typically used broad terms, while did not illustrate them in detail. Specifically, what specific distribution characteristics are important in modeling the population, and how do they affect optimization performance? Why should we capture these characteristics? Besides, "The reliance on approximate gradients during training can destabilize the model, hindering its overall effectiveness." Why does relying on approximate gradients lead to model instability? Are there theoretical or empirical studies supporting this claim? Providing concrete examples, intuitive explanations, or visual illustrations would help clarify the rationale behind the design choices.
3. Limited Novelty and Coherence. The issues addressed in this work (population modeling, information exchange, gradient estimation) have been studied individually in prior work, as the authors themselves acknowledge. While integrating these three aspects into a single framework could be valuable, the paper does not present strong evidence of novelty in either methodology or insight. Furthermore, the relationship among the three components is not well-articulated. The method appears as a collection of loosely related techniques rather than a cohesive, unified solution. The authors should elaborate on how these components interact and collectively contribute to solving zero-shot optimization.
4. The experiments are a little hard to follow. For example, it is difficult to understand the performance of various methods from Table 2. It also hard to identify the effectiveness of the proposed method compared with existing methods.

---

> ### Author Rebuttal · Authors · 2025-07-31
>
> > Weakness 1: Unclear Problem Definition...
>
> This article mainly addresses the zero-shot optimization problem for black-box optimization, focusing on its continuous optimization scenario. Based on the reviewer's suggestions, we have revised the second to fourth paragraphs of the Introduction to:
>
> Traditionally, solving BBO problems requires carefully designed or heavily tuned optimization algorithms for each individual task. This manual design process is costly, requires expert knowledge, and lacks scalability across diverse domains. To overcome these challenges, the paradigm of zero-shot optimization has emerged.
>
> Zero-shot optimization refers to the ability to solve a previously unseen optimization task directly—without any per-task tuning or adaptation. That is, given an unknown objective function $f$, the optimizer is expected to perform well without trial-and-error adjustments. This generalization ability is crucial for real-world applications where human-in-the-loop tuning is impractical.
>
> A promising approach to zero-shot optimization is the use of Pre-trained Optimization Models (POMs) \cite{li2024pretrained}, which learn general optimization strategies across a diverse training set of functions. POMs aim to transfer this knowledge to solve new BBO tasks efficiently and automatically.
>
> > W. 2: The motivation is unclear...
>
> Pre-trained optimization models (POMs) are a universal optimization framework designed to achieve zero-shot optimization. Their core idea is to learn the mapping between task features and optimization strategies, enabling the model to directly output high-quality optimization strategies without fine-tuning for new tasks.
>
> We summary this process into three key elements: task feature modeling, optimization strategies expression, and a mapping mechanism from task features to optimization strategies. First, the task feature set, typically consisting of normalized fitness values and centralized rankings of individuals, characterizes the structural characteristics of the optimization task. Its expressive power directly determines the POM's ability to perceive and adapt to different tasks. Second, the optimization strategies, as the output of the POM, must possess sufficient expressive power to cover search behaviors across diverse tasks. The mapping from features to strategies, learned during the pre-training phase, is key to achieving zero-shot performance.
>
> Although existing POM methods have achieved some success, significant bottlenecks remain in feature modeling, strategies expression, and training stability. To this end, this paper proposes systematic improvements around the three core elements mentioned above, building a POM system with enhanced generalization and training robustness:
>
> Enhanced Task Feature Modeling: To improve the expressiveness of task features, we propose the LMM/LCM Tokenizer. This module utilizes cross-attention to construct representations of decision variables of arbitrary dimensions and concatenates them with the original features, resulting in a richer and more structure-aware problem representation.
>
> Enhanced Optimization strategies Representation: We introduce deformable attention into the original LMM module, proposing the Efficient LMM, to enhance flexibility and information exchange during strategies generation, achieving significant performance improvements on multiple benchmark tasks.
>
> Enhanced Training Stability: To overcome the gradient estimation inaccuracies caused by Gumbel-Softmax, we design a differentiable crossover mechanism, Differentiable LCM. This replaces the original non-differentiable crossover process with a weighted summation, effectively improving the stability and convergence quality of the training process.
>
> In summary, by synergistically enhancing feature modeling, strategies representation, and mapping mechanisms, this paper promotes the development of pre-trained optimization models with stronger zero-shot optimization capabilities and wider adaptability.
>
> > W. 3: Limited Novelty...
>
> Please see our rebuttal to Weakness 2.
>
> > W. 4: The experiments are ...
>
> We hope that the following explanation of Table 2 can increase its readability.
> - Header $d$ means the problem dimension (30, 100), F means test problems (F1, F2, ..., F24), and EPOM, POM, ..., LGA are the evaluated methods.
> - Bottom $+/=/-: x/y/z$ shows that among the $24 \times 2$ tasks (24 BBOB functions in 2 dimension settings), EPOM is better than the comparison algorithm on $x$ tasks, comparable to the comparison algorithm on $y$ problems, and weaker than the comparison algorithm on $z$ problems.
> - Data For the data $a(b)$ in the table, $a$ represents the mean fitness of the last generation of population in the three average tests, and $b$ represents its standard deviation. The bold data represents the best among all methods, and the underlined data represents the second best or equally best.
>
> If there are any other issues regarding our Experiments section, we would be happy to discuss them.

---

> > ### Author Response · Authors · 2025-08-07
> >
> > We sincerely thank you for your initial review and the constructive comments. We have carefully addressed all the concerns in our rebuttal. As the discussion phase will conclude in 36 hours, we would greatly appreciate it if you could kindly take a moment to review our response and share any additional thoughts. Your input is invaluable to us, and we are grateful for your time and consideration.
> >
> > Sincerely,
> >
> > Authors

---

### Decision · Program_Chairs · 2025-09-17

**Decision:**

Accept (poster)

**Comment:**

This paper presents Efficient Pretrained Optimization Models (EPOM), a novel framework for zero-shot black-box optimization. The authors identify three key weaknesses in existing Pretrained Optimization Models (POMs)—poor population modeling, static solution interaction, and unstable gradient estimation—and introduce a series of well-motivated architectural improvements to address them. The core contributions include a new tokenizer that better captures landscape cues, a deformable attention module to promote solution diversity, and a differentiable crossover mechanism for more stable training. Together, these components create a more robust and efficient optimization model.

I recommend this paper for acceptance. It's a technically solid work that demonstrates strong empirical results on challenging benchmarks, clearly advancing the state of the art. The reviewers initially raised several important concerns, particularly regarding the clarity of the writing, motivation, and experimental results. However, the authors' thorough rebuttal successfully addressed these points, convincing all reviewers to converge on a positive recommendation. The paper tackles a significant problem, and its contributions are valuable to the optimization community, and I believe even more broadly to the NeurIPS community at large. I encouraged the authors to make sure they carefully integrate the excellent feedback from the reviewers and the discussion phase to further polish the paper for the camera-ready version. Great work!